# Linear Causal Representation Learning by Topological Ordering, Pruning, and Disentanglement

## Abstract

Causal representation learning (CRL) has garnered increasing interests from the causal inference and artificial intelligence community, due to its capability of disentangling potentially complex data-generating mechanism into causally interpretable latent features, by leveraging the heterogeneity of modern datasets. In this paper, we further contribute to the CRL literature, by focusing on the stylized linear structural causal model over the latent features and assuming a linear mixing function that maps latent features to the observed data or measurements. Existing linear CRL methods often rely on stringent assumptions, such as accessibility to single-node interventional data or restrictive distributional constraints on latent features and exogenous measurement noise. However, these prerequisites can be challenging to satisfy in certain scenarios. In this work, we propose a novel linear CRL algorithm that, unlike most existing linear CRL methods, operates under weaker assumptions about environment heterogeneity and data-generating distributions while still recovering latent causal features up to an equivalence class. We further validate our new algorithm via synthetic experiments and an interpretability analysis of large language models (LLMs), demonstrating both its superiority over competing methods in finite samples and its potential in integrating causality into AI. Source code is available at the anonymous link.

## 1 Introduction

How to organically integrate "causality" into modern artificial intelligence (AI) systems has become one of the central quests in the recent causal inference literature (Peters et al., 2017; Kıcıman et al., 2023; Lesci et al., 2024; Chen et al., 2024; Jørgensen et al., 2025). A fundamental principle along this direction is to leverage rich information from heterogeneous environments (i.e., datasets with heterogeneous distributions) (Dawid & Didelez, 2010; Yu, 2013; Bühlmann, 2020). Oriented by this principle, a particular promising strand of literature, called *causal representation learning* (CRL), has recently emerged (Schölkopf et al., 2021; Ahuja et al., 2023; Zhang et al., 2024a; Rajendran et al., 2024). Unlike tabular data encountered in social sciences, medicine, and epidemiology, in many modern scientific and industrial applications, measurements such as image pixels or language tokens often only contain low-level information of physically meaningful semantics.

The objective of CRL is then to uncover, from the low-level measurements lacking interpretable semantics, (1) the high-level, interpretable but latent features, and (2) the causal mechanism among the latent features. To the best of our knowledge, the term CRL was first coined in the important perspective paper by Schölkopf et al. (2021), although the essential idea can be traced back to factor analysis or independent component analysis (ICA) (Lawley & Maxwell, 1962; Comon, 1994; Hyvärinen & Oja, 2000) in the statistical literature.

Progress in CRL has advanced on several fronts since Schölkopf et al. (2021). For example, Ahuja et al. (2023) demonstrate that with hard interventional data, latent features can be identified up to shift and scaling transformations. The proposed approach learns latent features by optimizing a reconstruction-based objective function. In Buchholz et al. (2023), the difference in log-densities between the observational and interventional data is used as the loss function, by minimizing which one recovers the latent features and their causal mechanisms. A series of papers (Varıcı et al.,

2023; 2024b;a;c) establish that with linear mixing functions and multi-node interventions, hard interventions lead to perfect identifiability of latent causal representations, while soft interventions result in ancestral identifiability. For the nonlinear mixing scenario, similar identifiability and achievability are established assuming two hard or soft intervention per node. Zhang et al. (2024a) demonstrate that, under the sparsity constraint, latent features and causal mechanisms can be recovered as a function of itself and its neighbors in the Markov network implied by the ground truth causal graph. In a slightly different vein, having access to only observational data, Welch et al. (2024) show that latent features can be identified up to a layer-wise transformation consistent with the underlying causal-ordering and further disentanglement is impossible. Identifiability under linear CRL and in a more general setting was analyzed in Squires et al. (2023); Zhang et al. (2023).

In a seminal work, Jin & Syrgkanis (2024) conducted meticulous analysis of CRL by assuming that (1) latent features follow a linear structural causal model and (2) there exists a diffeomorphic linear mixing function that maps latent features to observed data. Linear CRL can still be relevant in practice based on recent work suggesting that there could be a linear relationship between the high-level, latent, but causally interpretable concepts and the last hidden states of large language models (LLMs) (Park et al., 2024; Arora et al., 2016; Mikolov et al., 2013); also see Section 4.2.

Jin & Syrgkanis (2024) also conducted *identification analysis* (under more general nonlinear models) without restricting to multiple environments generated from interventional data and propose an algorithm called `LiNGCReL`, recovering latent features and the underlying causal graph up to surrounded-node ambiguity (Varıcı et al., 2023). Their algorithm relies on several additional assumptions on exogenous noise variables, including (1) them being identically distributed across diverse environments and (2) their different components within the same environment having different distributions. However, the noise distributions across different environments can easily be different. For instance, data collected from different labs or experimentation equipments could have different types of noises, due to the heterogeneity in measurement devices. Yet noise components within one environment could be more likely to share the same distribution, because the measurements are presumably recorded using the same device or under a common environmental condition. The above reasoning motivates us to relax the assumptions imposed in Jin & Syrgkanis (2024) and develop a new linear CRL algorithm.

Our main contributions can be summarized as follows:

- We approach the linear CRL problem by relaxing some of the distributional assumptions required by the existing methods and assume only non-Gaussianity of the exogenous noise variables in the linear structural causal model; see Section 2. These assumptions, however, are critical for aligning the recovered exogenous noises across multiple environments, a key step in the algorithm proposed in Jin & Syrgkanis (2024); see Remark 2 for more details.

- We resolve these difficulties by designing a new CRL algorithm that can provably identify latent features and their causal mechanisms up to an equivalence class. The algorithm consists of three main subroutines: inferring the topological ordering, pruning, and finally disentangling latent features. In particular, we provide a necessary and sufficient condition for discovering latent exogenous noise as linear combinations of the observed variables (Theorem 2), and propose an iterative algorithm to infer the topological ordering of latent features based on this result. We will make these subroutines more precise in Section 3.

- We conduct synthetic experiments to evaluate the finite sample performance of our algorithm against `LiNGCReL`. We also apply our algorithm to the task of discovering latent causal features of LLMs output, demonstrating the practical utility of our new algorithm in helping us understand LLMs. See Section 4 for more details.

Our work is similar to Dong et al. (2024); Xie et al. (2024) in the sense that both investigate structure learning with latent variables. In Dong et al. (2024); Xie et al. (2024), the causal graph is recovered up to Markov Equivalence Class or perfectly recovered given certain additional assumptions on the graph structure, from observation data in one environment. However, our work leverages data from multiple environments and recovers the latent causal graph up to permutation without restriction on the graph structure. Moreover, our algorithm can recover both latent causal graph and latent feature up to some equivalence class while [1, 2] mainly focus on identifying the causal graph among observed and latent variables.

**Notation** Before moving forward, we collect some notation frequently used in later sections. For any natural number $n \in \mathbb{Z}_+$, we let $[n] := \{1, 2, \cdots, n\}$. The causal graph is denoted as $\mathcal{G} = \mathcal{G}(\mathcal{V}, \mathcal{E})$, where $\mathcal{V} := [d]$ is the set of $d$ nodes and $\mathcal{E}$ is the set of edges describing the causal relationship between nodes. We restrict $\mathcal{G}$ to be Directed Acyclic Graphs (DAGs).

We adopt the common familial terminologies in graphical models. For each node $i \in \mathcal{V}$, $\mathrm{pa}_{\mathcal{G}}(i)$ and $\mathrm{ch}_{\mathcal{G}}(i)$ denote, respectively, the parents and children of $i$ with respect to DAG $\mathcal{G}$. We follow the convention that each node is its own ancestor and descendant, adopted in earlier works in causal graphical models. We also let $\overline{\mathrm{pa}}_{\mathcal{G}}(i) := \mathrm{pa}_{\mathcal{G}}(i) \cup \{i\}$ and similarly $\overline{\mathrm{ch}}_{\mathcal{G}}(i) := \mathrm{ch}_{\mathcal{G}}(i) \cup \{i\}$. When it incurs no ambiguity, we silence the dependence on $\mathcal{G}$ and write, for instance, $\mathrm{pa}(i)$ instead of $\mathrm{pa}_{\mathcal{G}}(i)$. To all nodes $i \in \mathcal{V}$ correspond a vector of $d$ random variables $\{y_i \in \mathbb{R}, i \in \mathcal{V}\}$, whose joint probability distribution Markov factorizes with respect to (w.r.t.) $\mathcal{G}$. We use small letters $(x, y, \ldots)$ for one random variable/vector and reserve capital letters $(X, Y, \ldots)$ for $n$ i.i.d. copies of that random variable/vector. As in Jin & Syrgkanis (2024), we also introduce the surrounding set, defined as: for $i \in \mathcal{V}$, $\mathrm{sur}_{\mathcal{G}}(i) := \{j \in \mathcal{V} : j \in \mathrm{pa}_{\mathcal{G}}(i), \mathrm{ch}_{\mathcal{G}}(i) \subseteq \mathrm{ch}_{\mathcal{G}}(j)\}$, and $\overline{\mathrm{sur}}_{\mathcal{G}}(i) := \mathrm{sur}_{\mathcal{G}}(i) \cup \{i\}$. Besides, for any positive integer $m$, vector $x \in \mathbb{R}^m$, and subset $S \subset [m]$, we define $x_{i \in S}$ as vector $(x_i, i \in S)^\top$. Also, $\forall k \in [K], i \in [d] \setminus \{1\}$, we define $\mathrm{proj}_i^\perp x^{(k)} := x^{(k)} - \mathbb{E}(x^{(k)} | z_{j \in [i-1]})$ and $\mathrm{proj}_1^\perp x^{(k)} := x^{(k)}$, and similarly $\mathrm{proj}_i^\perp z^{(k)} := z^{(k)} - \mathbb{E}(z^{(k)} | z_{j \in [i-1]}^{(k)})$ and $\mathrm{proj}_1^\perp z^{(k)} := z^{(k)}$. As we focus on linear models, the conditional expectation reduces to population linear regression in this paper. Further clarifications of our notation in this paper are provided in Appendix A. Finally, for any matrix $A \in \mathbb{R}^{n_1 \times n_2}$, we denote its $i$-th row vector as $A_{i,\cdot}$ and $j$-th column vector as $A_{\cdot,j}$ and for any integer $i, j_1 \leq j_2 - 1$, $A_{i,j_1:j_2}$ represents the subvector of $A_{i,\cdot}$ from the $j_1$-th to $j_2$-th components.

## 2 PROBLEM SETUP AND IDENTIFIABILITY ANALYSIS

In this section, we describe the problem of linear CRL from heterogeneous environments, along with our assumptions, and an identifiability analysis. To set the stage, we assume that one has access to data collected from multiple environments $k \in [K]$. Different environments share the same set of "causal variables" denoted as $y^{(k)} \in \mathbb{R}^d$, governed by the same causal DAG $\mathcal{G}$. Different environments may differ in the joint probability distributions of $y^{(k)}$ for $k \in [K]$, which is also the reason why we attach a superscript to $y$. In our work, we assume that the latent dimension $d$ is known a priori. For scenarios where the latent dimension is unknown, established factor analysis methods (Onatski, 2010) can in principle be utilized to estimate the appropriate number of latent dimensions.

In CRL, $y^{(k)}$'s are latent, while the investigator instead gets to observe $p$-dimensional measurements $x^{(k)}$ in each environment, which served as proxies of the underlying causal variables $y^{(k)}$. In this paper, we assume that these proxies $x^{(k)}$ relate to $y^{(k)}$ through a linear mixing map $H : \mathbb{R}^d \to \mathbb{R}^p$ with $d \leq p$ invariant to $k \in [K]$:

$$y^{(k)} = W^{(k)\top} y^{(k)} + \Omega^{(k)} z^{(k)}, \quad x^{(k)} = H y^{(k)}, \tag{1}$$

where the matrix $W^{(k)} = (w_{i,j}^{(k)})_{i,j=1}^d$ is the weighted adjacency matrix of $\mathcal{G}$ satisfying that $w_{i,j}^{(k)} \neq 0$ if and only if $i$ is a parent node of $j$ in $\mathcal{G}$ and $\Omega^{(k)}$ is a diagonal matrix with positive entries. We let $X^{(k)} := (x_1^{(k)}, \cdots, x_n^{(k)})^\top$ denote the $n \times p$ data matrix gathering $n$ i.i.d. repeated draws of $x^{(k)}$ and similarly define $Y^{(k)} \in \mathbb{R}^{n \times d}$. Obviously, we have $X^{(k)} \equiv Y^{(k)} H$. The goal is to identify $y^{(k)}$ and $\mathcal{G}$ based on the observed data. However, just under the model defined via (1), it is not sufficient to identify latent features and the causal mechanisms $\mathcal{G}$ just based on $\mathbf{x} := \{x^{(k)}, k = 1, \cdots, K\}$. It is noteworthy that the linear mixing map $H$ and causal graph $\mathcal{G}$ are invariant across environments in our model. We always denote environment-dependent variables/matrices with superscripts, such as $z^{(k)}$ and $x^{(k)}$. The following additional assumptions are also imposed in this paper.

**Assumption 1.** *The exogenous noise $z^{(k)} \in \mathbb{R}^d$ has independent components; at most one component is Gaussian.*

**Assumption 2.** *The matrices $\{U^{(k)} := (\Omega^{(k)})^{-1}(I - W^{(k)})^\top\}$ are called node-level non-degenerate if for any node $i \in [d]$, $\dim \mathrm{span}\{U_{(i)}^{(k)} : k \in [K]\} = |\mathrm{pa}(i)| + 1$ where $U_{(i)}^{(k)}$ is the ith row of $U^{(k)}$.*

**Assumption 3.** *The mixing matrix $H \in \mathbb{R}^{n \times d}$ has full column rank.*

Assumption 1 imposes strictly weaker conditions on the exogenous noise than those in Jin & Syrgkanis (2024). In particular, it allows (1) the overall noise distribution to vary freely across environments, and

(2) permits each component of $z^{(k)}$ to follow any non-Gaussian distribution within each environment. Although these relaxed assumptions improve practical applicability, they require us to develop a new alignment procedure that matches noise components across environments — a step that remains essential in the method of Jin & Syrgkanis (2024), which instead assumes a common noise distribution across environments and heterogeneity only across components.

Assumptions 2–3 are adopted from Jin & Syrgkanis (2024). The central objective of Assumption 2 is to ensure sufficient heterogeneity across diverse environments, thereby enabling the identification of latent features and causal structures by exploiting variations among different environments. It guarantees that for every node $i$, the associated weight matrix, with each row corresponding to the weight vector $w_{i,\cdot}^{(k)}$ in one environment, is of column rank $|\mathrm{pa}(i)|+1$. Similarly, Assumption 3 ensures sufficient heterogeneity within and across environments.

Before proceeding, we make precise the meaning of identifying $y^{(k)}$ and $\mathcal{G}$ under Model (1), by further introducing the following definitions.

**Definition 1** (Equivalence up to permutation and scale). We write $\widehat{y}^{(k)} \sim_\pi y^{(k)}$ if there exists a permutation matrix $P_\pi$ corresponding to a permutation $\pi$ on $[d]$ and a non-singular diagonal matrix $\Gamma^{(k)}$ such that $y^{(k)} = P_\pi \Gamma^{(k)} \widehat{y}^{(k)}, \forall k \in [K]$. In words, $\widehat{y}^{(k)}$ and $y^{(k)}$ are equivalent up to permutation and scale.

**Definition 2** (Equivalence up to permutation after ordered linear transformation). We write $\widehat{y}^{(k)} \sim_\triangle y^{(k)}$ if there exists a permutation matrix $P_\pi$ and a lower triangular matrix $B$ such that $\widehat{y}^{(k)} = B P_\pi y^{(k)}, \forall k \in [K]$. In words, $\widehat{y}^{(k)}$ and $y^{(k)}$ are equivalent up to permutation after linear transformations based on a certain topological ordering.

**Definition 3.** We write $(\widehat{y}^{(k)}, \widehat{\mathcal{G}}) \sim_{\mathrm{sur}} (y^{(k)}, \mathcal{G})$ if $\forall k \in [K]$, there exists a permutation $\pi$ on $[d]$ and a lower triangular matrix $B$ where for $\forall j \in [d], i \notin \overline{\mathrm{sur}}(j)$, $B_{i,j} = 0$, such that the following holds:

- $\forall i, j \in [d], i \in \mathrm{pa}(j) \iff \pi(i) \in \mathrm{pa}(\pi(j))$;

- $\widehat{y}^{(k)} = B P_\pi y^{(k)}$, where $P_\pi$ denotes the permutation matrix corresponding to $\pi$.

Definition 3 was, to our knowledge, first considered in Jin & Syrgkanis (2024) as well. In our paper, when the recovered causal DAG $\widehat{\mathcal{G}}$ has already satisfied the restriction in Definition 3, we slightly abuse notation and write $\widehat{y}^{(k)} \sim_{\mathrm{sur}} y^{(k)}$ for short. To better illustrate the definition of $\sim_\pi$, $\sim_\triangle$ and $\sim_{\mathrm{sur}}$, we provide a three-node example in Appendix G.1.

The following theorem, proved in Appendix B, shows that the above assumptions ensure identifiability.

**Theorem 1.** *Under Assumptions 1–3, the distribution of the observed data $\{x^{(k)}, k \in K\}$ from at least $d$ environments identifies the latent features $\{y^{(k)}, k \in K\}$ and the true causal DAG $\mathcal{G}$ up to $\sim_{\mathrm{sur}}$.*

Before detailing our algorithm, we first clarify the scope of our theoretical contributions. Consistent with much of the current CRL literature, we focus on identifiability – namely, whether the latent features and causal DAG can be uniquely recovered in the limit of infinite data, and whether our proposed procedure achieves this recovery in principle. However, questions of statistical complexity lie outside the scope of this paper.

## 3 THE NEW LINEAR CRL ALGORITHM

In this section, we introduce CREATOR (Causal REpresentation leArning via Topological Ordering, Pruning, and Disentanglement), a novel linear CRL algorithm grounded in Theorem 1 and detailed in Algorithm 1. CREATOR proceeds in three *subroutines*:

1. **Topological Ordering & Feature Recovery.** Infer a causal ordering and recover latent features up to the equivalence relation $\sim_\triangle$.

2. **DAG Pruning.** Sparsify the initially dense DAG obtained in subroutine 1.

3. **Feature Disentanglement.** Refine latent features up to the equivalence relation $\sim_{\mathrm{sur}}$, leveraging the results of the first two subroutines.

## 3.1 SUBROUTINE 1: LATENT FEATURE LEARNING UP TO $\sim_\Delta$ BY INFERRING TOPOLOGICAL ORDERING

For simplicity, we fix the topological ordering as $\pi = (1, 2, \ldots, d)$. To learn latent features $y^{(k)}$, the first subroutine of CREATOR sequentially recovers one component $y_i^{(k)}$ and $z_i^{(k)}$ of $y^{(k)}$ and $z^{(k)}$ at a time, starting from the root/childless nodes. An illustration using $d = 3$ latent features is shown in Figure 1. As will be proved in Theorem 3, the order at which $y_i^{(k)}$ is recovered corresponds to its topological ordering encoded in the causal DAG $\mathcal{G}$.

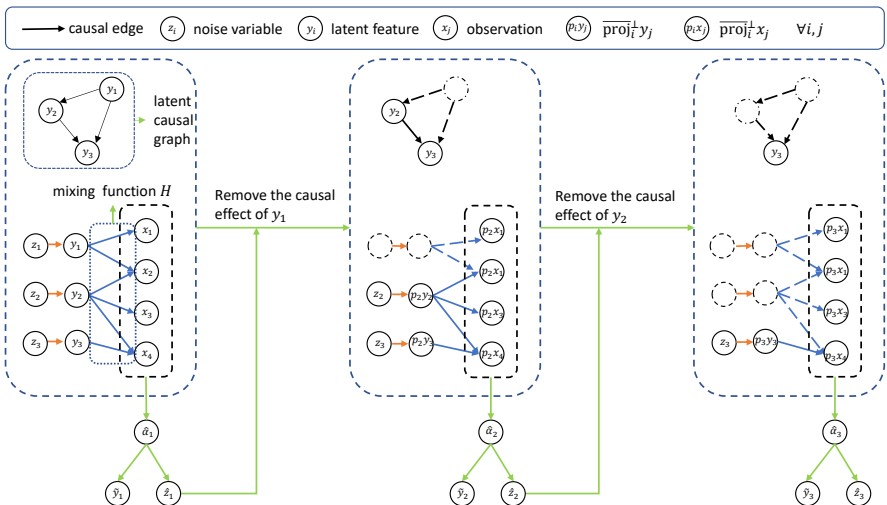

Figure 1: An illustration of subroutine 1. Dashed nodes and edges are eliminated.

Under Model (1) and Assumption 3, for any component $i \in [d]$, $y_i^{(k)} = \alpha_i^\top x^{(k)}$ for some $\alpha_i \in \mathbb{R}^p$. Therefore, we only need to obtain an appropriate $\alpha_i$ ($i \in [d]$) to recover $y^{(k)}$. The intuition of identifying a correct $\alpha$ can be gathered from Theorem 2 below.

**Theorem 2.** *Under Model* (1) *and Assumptions 1–3, for any nonzero $\alpha_i$ such that $\alpha_i^\top x^{(k)}$ is independent of $x^{(k)} - \mathbb{E}(x^{(k)}|\alpha_i^\top x^{(k)})$ for any $k \in [K]$, then $\alpha_i^\top x^{(k)} \propto_k z_i^{(k)}$ with i being a root node in $\mathcal{G}$ which implies $\alpha_i^\top x^{(k)} \propto_k y_i^{(k)}$, where $\propto_k$ means "equal up to some constant depending on k".*

We postpone the proof of Theorem 2 to Appendix D and only give a sketch here. For any $M^{(k)} \in \mathbb{R}^{n \times d}$ and $x^{(k)}$ generated by $x^{(k)} := M^{(k)} z^{(k)}$, by Darmois-Skitovitch theorem (Darmois, 1953; Skitovitch, 1953), $\alpha^\top x^{(k)}$ is independent of $x^{(k)} - \mathbb{E}(x^{(k)}|\alpha^\top x^{(k)})$ if and only if $\alpha^\top x^{(k)}$ is a component of $z^{(k)}$. Due to Model (1), $M^{(k)} = H(I - W^{(k)\top})^{-1}\Omega^{(k)}$, together with the acyclicity of $\mathcal{G}$, this component must correspond to one of the root nodes. Note that under the linear CRL Model (1), all conditional expectations appeared are linear combinations of the variables being conditioned upon.

To describe subroutine 1, we first explain our approach to discovering a root-node component of $y^{(k)}$. By Theorem 2, there exists $\alpha$ such that $\alpha^\top x^{(k)}$ corresponds to a root-node component of $y^{(k)}$ up to constant. We devise the following constrained optimization problem to identify such an $\alpha$, denoted as $\widehat{\alpha}$:

$$\widehat{\alpha}_i := \arg\min_\alpha L^{(i)}(\alpha, x) := \sum_{k=1}^K \sum_{j=1}^d \text{MI}(\alpha^\top \text{proj}_i^\perp x^{(k)}, r_j^{(k)}) \text{ s.t. } \alpha \in \bigcup_{k=1}^K \text{ica}(\text{proj}_i^\perp x^{(k)}), \quad (2)$$

where $r_j^{(k)} := \text{proj}_i^\perp x_j^{(k)} - \mathbb{E}(\text{proj}_i^\perp x_j^{(k)}|\alpha^\top \text{proj}_j^\perp x^{(k)})$, $\text{MI}(\xi, \eta)$ is the mutual information between two random variables $\xi$ and $\eta$, and $\text{ica}(\cdot)$ denotes the set of all row vectors of the unmixing matrices estimated by any ICA algorithm (Miettinen et al., 2015).

We now unpack (2). To ease exposition, we start by explaining the heuristic of the first iteration $i = 1$. As illustrated before Theorem 2, to identify $\alpha_1$ such that $\forall k \in [K]$, $\alpha_1^\top x^{(k)}$ equals to one of the

root-node components of $y^{(k)}$ is equivalent to finding $\alpha_1$ such that $\alpha_1^\top x^{(k)}$ and $r_j^{(k)}$ are independent for any $k \in [K]$. (2) achieves this by minimizing their mutual information. Since for any root node $j$, $y_j^{(k)} \equiv z_j^{(k)}$ for all $k \in [K]$, we only need to find $\alpha_1$ such that $\exists k_0 \in [K]$, such that $\alpha_1^\top x^{(k_0)}$ is a component of $z^{(k_0)}$. We leverage ICA to obtain unmixing matrices $N^{(k)}$ such that $N^{(k)} x^{(k)} = z^{(k)}$. Then $\forall j \in [d]$, we have $N_{j,\cdot}^{(k)\top} x^{(k)} = z_j^{(k)}$. Therefore, instead of directly solving the continuous optimization problem, we simply search over all $K \cdot d$ row vectors from all $K$ unmixing matrices $\{N^{(k)}, k \in [K]\}$ to identify $\widehat{\alpha}_1$. As we only need to identify $\widehat{\alpha}_1$ such that the mutual information in (2) is 0, we replace mutual information by independence criterion such as Hilbert-Schmidt Independence Criterion (HSIC) (Gretton et al., 2005). In turn, we obtain estimated version of $z_1^{(k)}$ and $y_1^{(k)}$, denoted as $\widehat{z}_1^{(k)} := \widehat{\alpha}_1^\top x^{(k)}$ and $\widetilde{y}_1^{(k)} := \widehat{\alpha}_1^\top x^{(k)}$ (see Remark 1 for why we use $\widetilde{y}^{(k)}$ instead of $\widehat{y}^{(k)}$).

Next, we obtain $\mathrm{proj}_2^\perp x^{(k)}$ by projecting $x^{(k)}$ onto the orthocomplement to $\widehat{z}_1^{(k)}$, by which the causal influences from $y_1^{(k)}$ to $y_j^{(k)}$ for $j \geq 2$ are eliminated. Graphically, this operation removes the first node and its connected edges in the original causal DAG $\mathcal{G}$. After this variable elimination process, new root nodes emerge so we can repeat this step iteratively to unravel the topological ordering of $\mathcal{G}$. A visual explanation can be found in Figure 1.

For iteration $i \geq 2$, by the definition of $\mathrm{proj}_i^\perp x^{(k)}$ and Model (1), we have $\mathrm{proj}_i^\perp x^{(k)} = H(I - W^{(k)\top})^{-1}\Omega^{(k)}\mathrm{proj}_i^\perp z^{(k)}$. As $\mathrm{proj}_i^\perp z_{j \in [i-1]}^{(k)} = 0$, $\mathrm{proj}_i^\perp x^{(k)}$ can only be a function of $z_j^{(k)}$, for $j \in \{i, i+1, \cdots, d\}$. As mentioned, we repeatedly solve (2) to obtain $\widehat{\alpha}_i$ and in turn the estimated $z_i^{(k)}$ and $y_i^{(k)}$, denoted as $\widehat{z}_i^{(k)} := \widehat{\alpha}_i^\top x^{(k)}$ and $\widetilde{y}_i^{(k)} := \widehat{\alpha}_i^\top x^{(k)}$, for $i = 1, \cdots, d$. We further define $\overline{\mathrm{proj}_i^\perp} x^{(k)} := x^{(k)} - \mathbb{E}(x^{(k)}|\widehat{z}_{j \in [i-1]})$ for $i \geq 2$ and $\overline{\mathrm{proj}_1^\perp} x^{(k)} := x^{(k)}$.

**Remark 1.** We use $\widetilde{y}^{(k)}$ instead of $\widehat{y}^{(k)}$ in subroutine 1 because further disentanglement for $\widetilde{y}^{(k)}$ is needed; the final estimator of $y^{(k)}$ is denoted as $\widehat{y}^{(k)}$. Recall that at iteration $i$, $\widetilde{y}_i^{(k)} = \widehat{\alpha}_i^\top x^{(k)} = \widehat{\alpha}_i^\top H y^{(k)}$ and $\widehat{\alpha}_i$ is the output of ICA. Let $\beta := \widehat{\alpha}_i^\top H$. We can only guarantee that $\beta_j \equiv 0$ for $j \in \{i+1, \ldots, d\}$, but not for $j \leq i$. Hence, $\widetilde{y}_i^{(k)}$ might depend on $y_j^{(k)}, \forall j \in [i-1]$, which is described informally as being "entangled" in this paper. The entanglement is equivalent to the matrix $B$ such that $\widetilde{y}^{(k)} = By^{(k)}$, where $B$ is a lower triangular matrix comprised of $\beta$ just defined. The procedure for disentangling $\widetilde{y}^{(k)}$ will be described in Section 3.3.

---

**Algorithm 1** CREATOR

---

**Input:** Observation data: $X := \{X^{(k)}, k \in [K]\}$
**Output:** estimated causal latent feature $\widehat{Y}^{(k)}$, latent causal graph $\widehat{\mathcal{G}}$

1: $\overline{\mathrm{proj}_1^\perp} X^{(k)} \leftarrow X^{(k)}$          $\triangledown$ subroutine 1
2: **for all** $i \in \{1, \cdots, d\}$ **do**
3:    $\widehat{\alpha}_i \leftarrow \arg\min_{\alpha} L^{(i)}(\alpha, X)$; $\widetilde{Y}_i^{(k)} \leftarrow X^{(k)}\widehat{\alpha}_i$; $\widehat{Z}_i^{(k)} \leftarrow \mathrm{proj}_i^\perp X^{(k)}\widehat{\alpha}_i$
4:    $\overline{\mathrm{proj}_{i+1}^\perp} X^{(k)} \leftarrow \overline{\mathrm{proj}_i^\perp} X^{(k)} - \mathbb{E}(\overline{\mathrm{proj}_i^\perp} X^{(k)}|\widehat{Z}_i)$
5: **end for**
6: $\widehat{\mathcal{G}} \leftarrow \mathtt{Pruning}(\widetilde{Y}^{(k)}, \widehat{Z}^{(k)})$        $\triangledown$ subroutine 2 (Section 3.2; Algorithm 2)
7: $\widehat{Y}^{(k)} \leftarrow \mathtt{Disentanglement}(\widetilde{Y}^{(k)}, \widehat{Z}^{(k)}, \widehat{\mathcal{G}})$    $\triangledown$ subroutine 3 (Section 3.3; Algorithm 3)

---

Owing to the above reasoning, we obtain the following theorem regarding the validity of subroutine 1. The proof is deferred to Appendix D.

**Theorem 3.** *Suppose that the optimization problem of subroutine 1 in Algorithm 1 is perfectly solved and denote the solution as $\widetilde{y}^{(k)}$ and $\widehat{z}^{(k)}$. Then we must have $\widehat{z}^{(k)} \sim_p z^{(k)}$ and $\widetilde{y}^{(k)} \sim_\triangle y^{(k)}$.*

**Remark 2.** In Jin & Syrgkanis (2024), extra distributional assumptions on $z^{(k)}$ are required to align the recovered exogenous noise variables across different environments. However, this is not necessary for us as they are automatically aligned by following the topological orderings.

## 3.2 Subroutine 2: Pruning

Subroutine 1 of `CREATOR` only identifies the topological ordering of $\mathcal{G}$, bearing extraneous edges. To further refine the causal DAG $\mathcal{G}$, we introduce a "pruning" subroutine as the second stage of `CREATOR`, which we now describe in detail. According to model (1), recovering the edges of $\mathcal{G}$ is equivalent to finding indices of nonzero elements in $W^{(k)}$. To obtain a proxy of $W^{(k)}$, we regress $\widehat{z}^{(k)}$ against $\widetilde{y}^{(k)}$ and denote the regression coefficient as $\widehat{B}^{(k)} \in \mathbb{R}^{d \times d}$. In the ideal case when $\widehat{z}^{(k)} \sim_p z^{(k)}$ and $\widetilde{y}^{(k)} \sim_\Delta y^{(k)}$, $\widehat{B}^{(k)} \equiv (\Omega^{(k)})^{-1}(I - W^{(k)})^\top B^{-1}$. For any different $1 \leq j \leq i - 1 \leq d$, $\widehat{B}_{i,j}^{(k)} = (\Omega^{(k)-1})_{\cdot,i}(B^{-1})_{\cdot,j}^\top(e_i - W_{\cdot,i}^{(k)})$. Then we construct $\mathbb{R}^K \ni \widehat{B}_{i,j} := (\widehat{B}_{i,j}^{(k)}, k \in [K])$ and $\mathbb{R}^{K \times (i-j)} \ni \widehat{C}_{i,j} := (\widehat{B}_{i,l}, l \in \{j+1, \ldots, i\})$. If $j \in \mathrm{pa}(i)$, $\widehat{B}_{i,j}^{(k)} = (\Omega^{(k)-1})_{\cdot,i}(B^{-1})_{\cdot,j}^\top(e_i - W_{\cdot,i}^{(k)})$ depends on $W_{j,i}^{(k)}$, while $\widehat{C}_{i,j}$ only depends on $W_{l,i}^{(k)}$ for $l \geq j+1$.

Thanks to the heterogeneity across environments as imposed in Assumption 2, $\widehat{B}_{i,j}$ cannot be expressed as a linear combination of column vectors of $\widehat{C}_{i,j}$, which further implies that the rank of $\widehat{C}_{i,j}$ must be less than that of $[\widehat{C}_{i,j}, \widehat{B}_{i,j}]$. The pruning step, the pseudocode of which can be found in Algorithm 2 in Appendix C, essentially leverages this rank difference to remove spurious edges by iterating over all $\{(i, j), j < i\}$ pairs based on the inferred topological ordering from subroutine 1. We summarize the above reasoning in Theorem 4, with the complete proof deferred to Appendix D.

**Theorem 4.** *Under Model* (1) *and Assumptions 1–3*, $j \in \mathrm{pa}(i)$ *if and only if* $\mathrm{rank}(\widehat{C}_{i,j}) = \mathrm{rank}(\widetilde{C}_{i,j}) - 1$, *where* $\widetilde{C}_{i,j} := (\widehat{B}_{i,j}, \widehat{C}_{i,j})$.

We prune spurious edges using the estimate $\widehat{B}^{(k)} := (\Omega^{(k)})^{-1}(I - W^{(k)})^\top B^{-1}$. Given the topological ordering, at step $i$ we consider each candidate edge from node $j \leq i - 1$ to $i$. For each pair $(i, j)$, the columns of matrices $\widehat{C}_{i,j} \in \mathbb{R}^{K \times (i-j)}$ and $\widetilde{C}_{i,j} \in \mathbb{R}^{K \times (i-j+1)}$ are formed by the vectors $\widehat{B}_{i,\, j:i}^{(k)}$ for $k = 1, \ldots, K$. We remove the edge $j \to i$ if $\mathrm{rank}(\widetilde{C}_{i,j}) = \mathrm{rank}(\widehat{C}_{i,j})$. By contrast, Jin & Syrgkanis (2024) use an ICA unmixing matrix to select parents among all ancestors: they compute the dimension $r_i$ of the subspace spanned by the unmixing-matrix rows projected onto the orthogonal complement of the first $j - 1$ ancestor rows, and retain $j \to i$ only if $r_i = r_{i-1} - 1$. Our use of the inferred topological ordering reduces the dimensions of the matrices whose ranks must be evaluated, yielding a more efficient pruning procedure.

## 3.3 Subroutine 3: Feature Disentanglement

With the causal DAG and entangled latent features from the previous steps, we can disentangle latent features further up to the equivalence class $\sim_{\mathrm{sur}}$ using subroutine 3, the disentanglement algorithm (Algorithm 3 in Appendix C), with the pseudocode deferred to Appendix C. Since $\widetilde{Y}^{(k)} = BY^{(k)}$, for any $i \in [d]$, we need to learn the $i$-th row of $B^{-1}$ to disentangle $\widetilde{Y}_k$. As $B^{(k)} = (\Omega^{(k)})^{-1}(I - W^{(k)})^\top B^{-1}$, the row space spanned by $\{B_{j,\cdot}^{(k)}, j \in \overline{\mathrm{ch}}(i)\}$ is comprised of vectors $\breve{B}_i$ formed by linear combinations of $\{(B^{-1})_{j,\cdot}, j \in \mathrm{sur}(i)\}$. Let $\widehat{y}^{(k)} := \breve{B}_i^\top \widetilde{y}^{(k)}$, which is a linear combination of $y_{j \in \overline{\mathrm{sur}}(i)}^{(k)}$. Then by definition of $\sim_{\mathrm{sur}}$, we succeed in disentangling $\widetilde{y}^{(k)}$ into $\widehat{y}^{(k)} \sim_{\mathrm{sur}} y^{(k)}$. These arguments culminate at the following theorem, the proof of which is provided in Appendix D.

**Theorem 5.** *Let* $\widehat{y}^{(k)}$ *and* $\widehat{\mathcal{G}}$ *for* $k \in [K]$, *be the solutions returned by Algorithm 3 and Algorithm 2. Under Model* (1) *and Assumptions* $1 - 3$, *we have* $(\widehat{y}^{(k)}, \widehat{\mathcal{G}}) \sim_{\mathrm{sur}} (y^{(k)}, \mathcal{G})$ *for all* $k \in [K]$.

**Statistical Complexity** All previous results concern whether `CREATOR` can identify the latent features and the underlying causal DAG. Theorem 3 in Appendix D.1 further establishes the point-wise consistency of `CREATOR`, by proving that the latent features and the underlying causal DAG can be asymptotically recovered up to $\sim_{\mathrm{sur}}$ equivalence when $n \to \infty$. An estimate of the computational complexity of `CREATOR` can be found in Appendix D.2.

In Appendix G.2, we provide a toy example for Algorithm 1.

## 4 NUMERICAL EXPERIMENTS

### 4.1 SYNTHETIC EXPERIMENTS

In this section, we examine the finite sample performance of `CREATOR` against the method developed in Jin & Syrgkanis (2024) using synthetic experiments. As mentioned, several other studies with different settings about the data generation process from our work, notably Varıcı et al. (2024b;a). Since the setting considered here is more closely related to Jin & Syrgkanis (2024), we will only compare `CREATOR` with their algorithm `LiNGCReL` below.

**Experimental setup.** As in Model (1), we first generate the weighted adjacency matrices $W^{(k)}$ and the exogenous noise $Z^{(k)}$. The matrix $W^{(k)}$ is obtained by multiplying the binary adjacency matrix of the causal DAG $\mathcal{G}$ with a random weight matrix from various distributions. The causal DAG $\mathcal{G}$ is constructed based on the Erdős-Rényi random graph model (Erdős & Rényi, 1959). The matrix $Z$ is generated by sampling from a non-Gaussian distribution. More details can be found in Appendix E.2.

We evaluate `CREATOR` across various settings to assess its performance. In setting (1), we allow different noise distributions across different environments, without imposing further distributional assumptions on each component within a single environment, corresponding to the more relaxed assumptions considered in this paper. In setting (2), similar to Jin & Syrgkanis (2024), the noise distributions are invariant across environments, but the distributions between different components differ. In Appendix E.3, we also generate $W^{(k)}$ in the same procedure but multiply them by $\sigma \in \{0.005, 0.007, 0.01, 0.03, 0.05, 0.07, 0.1, 0.3, 0.5\}$, resulting in data with weak causal influences. In this case, the topological ordering inference is subject to substantial error. We demonstrate that inferring topological ordering is crucial for correctly extracting latent features. We use the structural Hamming distance (SHD) (smaller is better) to compare DAGs and a metric called LocR$^2$ (larger is better) to quantify the similarity between the learned and true latent features. The definitions of both metrics are deferred to Appendix E.1.

**Results** We randomly sample 50 causal models with latent feature dimension $d = 2, 3, 5, 7$ and for each $d$, we sample $K \in \{d, 2d\}$ environments each with sample size $n = 1000$. We compare `CREATOR` and `LiNGCReL` for different $d$ and $K = d$ and present the accuracy of learning the causal DAG and latent features in Figure 2. We present similar results in the same setting but with $K = 2d$ in Appendix E.2. From these figures, we observe that `CREATOR` performs better in LocR$^2$ and SHD for different dimensions in both settings.

### 4.2 LATENT CAUSAL MECHANISMS OF LLMS: A CASE STUDY

The working mechanism of LLMs has been an open problem in modern AI that attracts much attention (Bubeck et al., 2023; Allen-Zhu & Li, 2023). Several recent works report that high-level interpretable concepts encoded by LLMs might be linearly related (Park et al., 2024; Arora et al., 2016; Mikolov et al., 2013). Here we adopt this "linear representation hypothesis" and use CRL to study latent causal mechanisms of LLMs. Specifically, we generate three ($K = 3$) types of stories with sufficiently heterogeneous styles via GPT-4 (Achiam et al., 2023) and DeepSeek (Liu et al., 2024), including news ($k = 1$), fairy tales ($k = 2$), and plain texts ($k = 3$). Each story consists of three main parts: background (BG), condition (CD) and ending (ED), which are treated as latent causally interpretable features. By common sense, the causal DAG of these features should contain three edges: BG $\rightarrow$ CD, CD $\rightarrow$ ED and BG $\rightarrow$ ED. We input the generated stories to various LLMs and extract the last hidden states of the chosen LLMs as the observed data, denoted as $x^{(k)}, k \in [3]$.

Under the "linear representation hypothesis", we assume that each observation $x^{(k)}$ is a linear transformation of the high-level representations of a story's background (BG), condition (CD), and ending (ED). We then apply `CREATOR` and `LiNGCReL` (Jin & Syrgkanis, 2024) to infer the latent features $\widehat{y}^{(k)}$ and reconstruct the causal DAG $\widehat{\mathcal{G}}$.

Because true latent features are unavailable in real data, we query large language models during story generation to extract keywords for BG, CD, and ED as proxy ground truth (see Appendix E.4 for generation and query details). Since both latent features and DAGs are identifiable only up to the equivalence $\sim_{\mathrm{sur}}$, we find the permutation over latent features that minimizes the error of predicting latent features with proxies (via training a neural net) as the latent features with meaningful ordering.

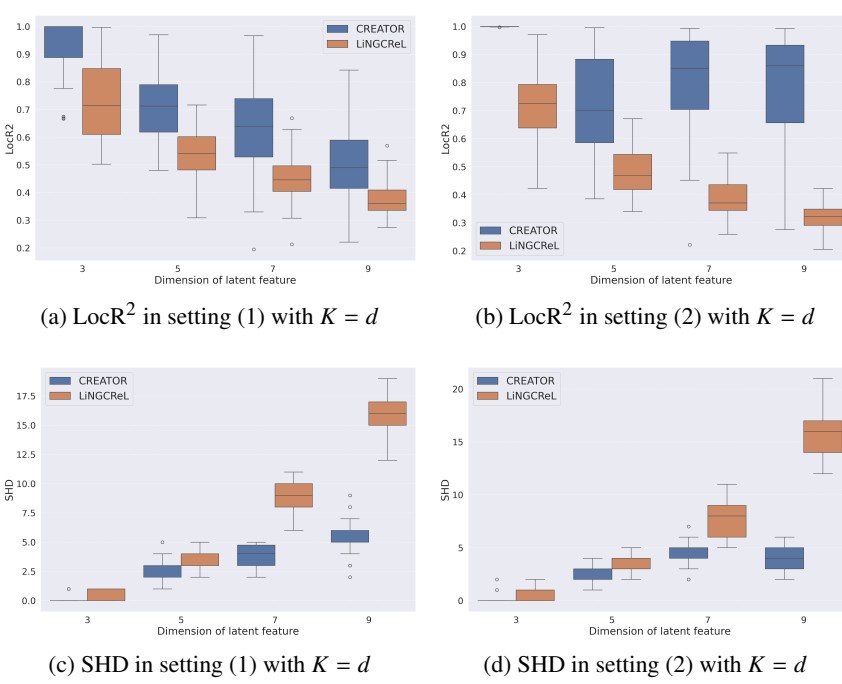

(a) LocR$^2$ in setting (1) with $K = d$      (b) LocR$^2$ in setting (2) with $K = d$

(c) SHD in setting (1) with $K = d$      (d) SHD in setting (2) with $K = d$

Figure 2: LocR$^2$ and SHD metric for different data generation setup. Figure 2a and 2c compare the performance of latent feature and causal DAG identification in setting (1). Figure 2b and 2d present performance in setting (2).

We evaluate each method by comparing its estimated DAG against the proxy ground truth; results are shown in Table 1. In this experiment, CREATOR recovers the causal structure more accurately than LiNGCReL, though broader evaluation is needed before making definitive recommendations. Further details are provided in Appendix E.4.

We remark that this case study only serves as a proof-of-concept exercise of CREATOR. Investigating how to apply or modify CRL methods like CREATOR to real-life problems, such as improving LLM interpretability and relevance in biological problems (Gao et al., 2025), is a natural next step of our work.

Table 1: Inferring latent causal mechanism of LLMs. ✓: correct DAG; ✕: incorrect DAG. Analyzed LLMs: Llama Guard (Llama Team, 2024), Llama 3.1 Instruct, TinyLlama (Zhang et al., 2024b), Phi-3-Mini (Abdin et al., 2024), GPT-Neo (Black et al., 2021), BLOOM (Scao et al., 2022).

| LLM | CREATOR | | | LiNGCReL | | |
|---|---|---|---|---|---|---|
| | BG → CD | CD → ED | BG → ED | BG → CD | CD → ED | BG → ED |
| Llama Guard | ✓ | ✓ | ✓ | ✓ | ✓ | ✓ |
| Llama 3.1 Instruct | ✓ | ✓ | ✓ | ✓ | ✕ | ✕ |
| TinyLlama | ✓ | ✓ | ✓ | ✕ | ✓ | ✓ |
| Phi-3-Mini | ✕ | ✓ | ✓ | ✕ | ✓ | ✕ |
| GPT-Neo | ✕ | ✓ | ✓ | ✕ | ✓ | ✕ |
| BLOOM | ✓ | ✕ | ✓ | ✕ | ✕ | ✕ |

## 5 CONCLUSIONS

In this paper, we present a new linear CRL algorithm under weaker assumptions, called CREATOR. By conducting numerical experiments, the algorithm demonstrates competitive performance in various

settings. We also apply CREATOR to uncover the latent causal mechanism of LLMs in a simplified setup, as a proof-of-concept of its potential value for the AI community.

There are several promising future directions. First, it is important to extend our algorithm to nonlinear settings (Varıcı et al., 2024a). In Rajendran et al. (2024), the problem of recovering latent causal structures is relaxed to recovering latent concepts, which leads to the possibility of requiring much fewer environments, as datasets can be quite costly to collect in many applications. This could be an interesting direction to explore and understand to what extent our algorithm CREATOR can also be used to recover "latent concepts".

We believe that evaluating the performance of CRL algorithms is an underexplored research area in the CRL literature. In real life applications, due to the unavailability of ground truth and the computational difficulty of evaluation even if ground truth is given, a reasonable approach is to evaluate the performance of downstream tasks of CRL [1]. For example, an important application of CRL is to leverage the underlying causal graph structure and intervene latent features to achieve certain desired changes in the observed data (such as natural images, sounds, and etc.) [2]. Whether the desired changes manifest after intervening the latent features can also serve as an indirect evidence of the success of CRL algorithms.

Besides, extending our algorithm to nonlinear situation is also very important and thus a discussion is provided in Appendix F.

Finally, it is also of interest to develop CRL algorithms that can handle dynamical data and multimodal data, as considered in several recent work on CRL (Zhang et al., 2024a; Song et al., 2024; Sun et al., 2025).

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

## A  FURTHER CLARIFICATION OF OUR NOTATION

In this section, we further explain some notations in this paper and provide definitions for the notations used but not mentioned in Section 1.

In this paper, the symbol $\mathbb{E}(\cdot|\cdot)$ is often used. For any two $d$-dimensional random variables $\xi$ and $\eta$, $\mathbb{E}(\xi|\eta)$ is the linear projection of $\xi$ onto the space spanned by $\eta$ under Model (1). To be concrete, $\mathbb{E}(\xi|\eta) = B^\top \eta \equiv \mathbb{E}[\xi\eta^\top](\mathbb{E}[\eta\eta^\top])^{-1}\eta$, where the true population regression coefficient term corresponds to the $d \times d$-dimensional matrix $B \coloneqq (\mathbb{E}[\eta\eta^\top])^{-1}\mathbb{E}[\eta\xi^\top]$. We denote their $n \times d$ sample matrices as $\underline{\xi} \coloneqq (\xi_1, \xi_2, \ldots, \xi_n)^\top$ and $\underline{\eta} \coloneqq (\eta_1, \eta_2, \ldots, \eta_n)^\top$, where $n$ is the sample size. In the actual implementation of our algorithm CREATOR, we estimate $\mathbb{E}(\xi|\eta)$ by $(\underline{\xi}^\top\underline{\eta})(\underline{\eta}^\top\underline{\eta})^{-1}\eta$.

Given any matrix $A \in \mathbb{R}^{n_1 \times n_2}$ and any indices $i_1, i_2 \in [n_1]$, $j_1, j_2 \in [n_2]$ satisfying $i_1 \leq i_2 - 1$ and $j_1 \leq j_2 - 1$, and index sets $S_1 \subseteq [n_1]$ and $S_2 \subseteq [n_2]$, we adopt the following submatrix and subvector notations:

- $A_{i_1, j_1:j_2}$ denotes the subvector of row $A_{i_1,\cdot}$ ranging from the $j_1$-th to $j_2$-th components;
- $A_{i_1:i_2, j_1}$ denotes the subvector of column $A_{\cdot, j_1}$ ranging from the $i_1$-th to $i_2$-th components;
- $A_{i_1:i_2, j_1:j_2}$ denotes the submatrix of $A$ formed by columns $A_{\cdot, j_1}$ through $A_{\cdot, j_2}$ and rows $A_{i_1,\cdot}$ through $A_{i_2,\cdot}$, specifically $(A_{i_1:i_2, j_1}, A_{i_1:i_2, j_1+1}, \ldots, A_{i_1:i_2, j_2})$;
- $A_{S_1, j_1}$ denotes the subvector of $A_{\cdot, j_1}$ corresponding to row indices in $S_1$;
- $A_{i_1, S_2}$ denotes the subvector of $A_{i_1,\cdot}$ corresponding to column indices in $S_2$;
- $A_{S_1, S_2}$ denotes the submatrix of $A$ with row indices in $S_1$ and column indices in $S_2$.

For any integer $k$, $\mathbf{1}_k$ denotes a vector in $\mathbb{R}^k$ with all entries being 1. Finally, we remark that throughout the paper we have used "estimated" frequently. Here "estimated" should be mainly interpreted as the output of our proposed algorithm from infinitely amount of observed data (or equivalently the observed-data distribution), as we have mostly focused on the identifiability issue. In synthetic experiments or real data analysis, we instead recover the latent features and causal DAG from finite samples, which truly corresponds to the usual meaning of "estimated" in statistics.

## B  PROOF OF THEOREM 1

We commence the proof by stating the following lemma.

**Lemma 1.** *For $i, j \in [d]$ with $i \neq j$, there does not exist $k_1, k_2 \in [K]$ with $k_1 \neq k_2$, such that $(I - W^{(k_1)})_{\cdot,i} \propto (I - W^{(k_2)})_{\cdot,j}$.*

*Proof.* For any two nodes $i$ and $j$, $(I - W^{(k_1)})_{i,i} = (I - W^{(k_2)})_{j,j} = 1$ because a DAG $\mathcal{G}$ cannot have self-cycles. Suppose that on the contrary, $(I - W^{(k_1)})_{\cdot,i} \propto (I - W^{(k_2)})_{\cdot,j}$. Recall that this notation means that $\exists \theta \in \mathbb{R}$ such that $(I - W^{(k_1)})_{\cdot,i} = \theta(I - W^{(k_2)})_{\cdot,j}$. Since there exist $W^{(k_1)})_{i,i} \neq 0$ and $(I - W^{(k_2)})_{j,j} \neq 0$, the constant $\theta \not\equiv 0$, implying that $(I - W^{(k_1)})_{j,i}$ and $(I - W^{(k_2)})_{i,j}$ must be nonzero as well. It in turn follows that $j \in \mathrm{pa}(i)$ and $i \in \mathrm{pa}(j)$, which violates the acyclicity of $\mathcal{G}$, a contradiction. $\square$

**Lemma 2.** *For any integer $d$, any $d$-dimensional diagonal matrix $\Omega$ with nonzero diagonal entries, and any permutation matrix $P \in \mathbb{R}^{d \times d}$, we have $P\Omega = \Omega^P P$ where $\Omega^P$ denotes the diagonal matrix whose diagonal entries are permuted from the diagonal entries of $\Omega$ by the permutation matrix $P$.*

*Proof.* By definition, $P\Omega P^\top = \Omega^P$, so we immediately have $\Omega^P P = P\Omega$. $\square$

Armed with Lemma 1 and Lemma 2, we prove Theorem 1 below.

*Proof of Theorem 1.* For simplicity and clarity, we first consider the case of $p = d$, and defer the generalization to the case $p \geq d + 1$ to the end of the proof. Let $(\widehat{z}^{(k)}, \widehat{\Omega}^{(k)}, \widehat{W}^{(k)}, \widehat{y}^{(k)}, \widehat{H})$ be any candidate solution that also satisfies the data generating model (1). By classical results on ICA (Comon, 1994; Hyvärinen & Oja, 2000), given the observed data $x^{(k)}$, generated by invertible linear mapping from non-Gaussian exogenous variables $z^{(k)}$ with independent components, $z^{(k)}$ could be recovered up to permutation and scaling transformations. Therefore, there exists a permutation matrix $P^{(k)}$ such that $\widehat{z}^{(k)} = \Gamma^{(k)} P^{(k)} z^{(k)}$ for any $k \in [K]$, where $\Gamma^{(k)}$ is a nonsingular diagonal matrix. Together with (1), we have

$$H(I - W^{(k)\top})^{-1}\Omega^{(k)} = \widehat{H}(I - \widehat{W}^{(k)\top})^{-1}\widehat{\Omega}^{(k)}\Gamma^{(k)}P^{(k)}$$

$$\Rightarrow \widehat{H}^{-1}H(I - W^{(k)\top})^{-1} = (I - \widehat{W}^{(k)\top})^{-1}\widehat{\Omega}^{(k)}\Gamma^{(k)}P^{(k)}(\Omega^{(k)})^{-1}$$

$$\Rightarrow (I - W^{(k)\top})H^{-1}\widehat{H} = \Omega^{(k)}(\Gamma^{(k)}P^{(k)})^{-1}(\widehat{\Omega}^{(k)})^{-1}(I - \widehat{W}^{(k)\top}),$$

By Lemma 2, we have $\Omega^{(k)}P^{(k)\top} = P^{(k)\top}\Omega^{(k)\dagger}$, where use $\Omega^{(k)\dagger}$ to denote $(\Omega^{(k)})^{P^{(k)}}$ to avoid notation clutter. By letting $T := H^{-1}\widehat{H}$ and $\widehat{\Omega}'^{(k)} := \Omega^{(k)\dagger}(\Gamma^{(k)})^{-1}(\widehat{\Omega}^{(k)})^{-1}$, we finally obtain that

$$(I - W^{(k)\top})T = P^{(k)\top}\Omega'^{(k)}(I - \widehat{W}^{(k)\top}). \tag{3}$$

Then we have, also by Lemma 2, $\forall k \in [K]$, $(I - W^{(k)\top})T \equiv \Omega'^{(k)}P^{(k)\top}(I - \widehat{W}^{(k)\top})$. Without loss of generality, let $I - W^{(k)\top}$ be a lower triangular matrix. Hence, the first row of $(I - W^{(k)\top})T$ reduces to $T_{1,\cdot} \equiv \Omega'^{(k)}_{1,1}P^{(k)\top}_{1,\cdot}(I - \widehat{W}^{(k)\top})$. As $P^{(k)}$ is a permutation matrix, denote the index of the nonzero entry of $P^{(k)}_{1,\cdot}$ as $p_{1,k}$, we obtain the identity $T_{1,\cdot} = \Omega'^{(k)}_{1,1}P^{(k)}_{1,\cdot}(I - \widehat{W}^{(k)\top}) = \Omega'^{(k)}_{1,1}(I - \widehat{W}^{(k)\top})_{p_{1,k},\cdot}$. We then conclude that the indices of nonzero entries of $T_{1,\cdot}$ correspond to $\mathrm{pa}_{\widehat{\mathcal{G}}}(p_{1,k})$. Since $T_{1,\cdot}$ does not vary with $k$, by Lemma 1, we can conclude that $p_{1,1} = p_{1,2} = \cdots = p_{1,K}$. As $\Omega'^{(k)}_{1,1}(I - \widehat{W}^{(k)\top})_{p_{1,k},p_{1,k}} = T_{1,p_{1,k}}$, we also have $\Omega'^{(1)}_{1,1} = \Omega'^{(2)}_{1,1} = \cdots = \Omega'^{(K)}_{1,1}$. It follows that $(I - \widehat{W}^{(k)\top})_{p_{1,k},\cdot} = e_{p_{1,k}}$, where $e_{p_{1,k}}$ is a vector with the $p_{1,k}$-th entry being 1 and other entries being 0. Without loss of generality, we suppose $p_{1,k} \equiv 1$ for all $k$. If this does not hold, we only need to swap node $p_{1,k}$ and node 1 in $\widehat{\mathcal{G}}$ for $k \in [K]$.

For the second row of $(I - W^{(k)\top})T$, we only need to consider the subvector from the second entry onward. Then we have $T_{2,2:d} = (\Omega'^{(k)}_{2,2}P^{(k)}_{2,\cdot}(I - \widehat{W}^{(k)\top}))_{2:d}$. Denote the index of the nonzero entry of $P^{(k)}_{2,\cdot}$ as $p_{2,k}$ and we obtain $T_{2,2:d} = \Omega'^{(k)}_{2,2}(I - \widehat{W}^{(k)\top}))_{p_{2,k},2:d}$. With a similar argument, we have $\Omega'^{(1)}_{2,2} = \Omega'^{(2)}_{2,2} = \cdots = \Omega'^{(K)}_{2,2}$. We now consider a subgraph $\widehat{\mathcal{G}}'$ of $\widehat{\mathcal{G}}$ with the first node

removed and denote its corresponding weighted adjacency matrix as $\widehat{W}'^{(k)} := \widehat{W}^{(k)}_{2:d,2:d}$. Then we have $T_{2,2:d} = \Omega'^{(k)}_{2,2}(I_{p_{2,k}} - \widehat{W}'^{(k)}_{p_{2,k}})$. By Lemma 1, we again have $p_{2,1} = p_{2,2} = \cdots = p_{2,K}$. Again, without loss of generality, we can take $p_{2,k} \equiv 2$ for all $k$. By repeatedly applying the above arguments, we can show that $\mathcal{G}$ and $\widehat{\mathcal{G}}$ are isomorphic and $P^{(1)} = P^{(2)} = \cdots = P^{(K)}$.

Up to this point, we are only left to prove $\forall k \in [K]$, $\widehat{y}^{(k)} \sim_{\text{sur}} y^{(k)}$. Now we finish the remaining part of the proof[1]. For simplicity, we suppose that $P = I$ and $\Omega'^{(k)} = I$ with no loss of generality, as our identifiability analysis is up to permutation and scaling transformations. Under this simplification, we have $I - W^{(k)\top} = (I - \widehat{W}^{(k)\top})T^{-1}$ and $\widehat{y}^{(k)} = T^{-1}y^{(k)}$ for all $k \in [K]$. For any two nodes $i, j \in [d]$ such that $i \notin \overline{\text{pa}}(j)$, we have $\forall k \in [K]$, $(I - W^{(k)\top})_{j,i} = 0$ and $(I - \widehat{W}^{(k)\top})_{j,i} = 0$. From the previous identity $I - W^{(k)\top} = (I - \widehat{W}^{(k)\top})T^{-1}$, we have $\sum_{l \in \overline{\text{pa}}(j)} (I - \widehat{W}^{(k)})_{l,j}(T^{-1})_{l,i} = 0$. By Assumption 2, $\forall l \in \overline{\text{pa}}(j)$, $(T^{-1})_{l,i} = 0$, implying that for any two different nodes $l, i \in [d]$, only if $\overline{\text{ch}}(l) \subseteq \overline{\text{ch}}(i)$, $(T^{-1})_{l,i} \neq 0$. Therefore, for any node $l \in [d]$, $\widehat{y}^{(k)}_l = \sum_{i \in [d]}(T^{-1})_{l,i}y^{(k)}_i = \sum_{i \in \overline{\text{sur}}(l)}(T^{-1})_{l,i}y^{(k)}_i$,

following that $(y^{(k)}, \mathcal{G}) \sim_{\text{sur}} (\widehat{y}^{(k)}, \widehat{\mathcal{G}})$.

For the scenario of $p \geq d + 1$, we can simply consider the first $d$ dimension of the observed data $x^{(k)}_{[d]}$ for $k \in [K]$, generated by $x^{(k)}_{[d]} = H_{[d],\cdot}y^{(k)} = H_{[d],\cdot}(I - W^{(k)\top})^{-1}\Omega^{(k)}z^{(k)}$, for any $k \in [K]$, and thus the identifiability of $(y^{(k)}, \mathcal{G})$ could be obtained by simply applying the same argument for $p = d$ to $\{x^{(k)}_{[d]}, k \in [K]\}$. $\qquad\square$

## C    Pseudocode for subroutines 2 and 3 in Algorithm 1

In this section, we document Algorithm 2 and Algorithm 3 mentioned in the main text. In Algorithm 2, we estimate the rank using singular value-threholds.

---

**Algorithm 2** Pruning

---

**Input:** data $X^{(k)}$, latent features $\widetilde{Y}^{(k)}$, noise $\widehat{Z}^{(k)}$ for $k \in [K]$ from subroutine 1
**Output:** pruned causal DAG $\mathcal{G}$

1: Denote adjacency matrix of causal DAG as $W$ with $W_{ij} = 1 \; \forall i < j$ and other elements 0.
2: **for all** $k \in \{1, \ldots, K\}$ **do**
3:     regress $\widehat{Z}^{(k)}$ on $\widetilde{Y}^{(k)}$ and denote the regression term as $\widehat{B}^{(k)}$
4: **end for**
5: **for all** $i \in \{2, \ldots, d\}$ **do**
6:     **for all** $j \in \{1, \ldots, i\}$ **do**
7:        Denote $\widehat{B}_{i,j} := (\widehat{B}^{(k)}_{i,j})_{k \in [K]}$ and $\widehat{C}_{i,j} := \widehat{B}_{i,j+1:i}$
8:        **if** $\text{rank}(\widehat{C}_{i,j}) = \text{rank}([\widehat{C}_{i,j}, \widehat{B}_{i,j}]) - 1$ **then**
9:          $W_{j,i} = 1$
10:       **else**
11:          $W_{j,i} = 0$
12:       **end if**
13:     **end for**
14: **end for**
15: Construct estimated causal DAG $\widehat{\mathcal{G}} = (\widehat{V}, \widehat{E})$ with $W$

---

**Remark 3.** In both the purning subroutine in our manuscript and the Identify-Parents algorithm in Jin & Syrgkanis (2024), SVD is conducted. In Jin & Syrgkanis (2024), dimension of space spanned by $K$ vectors in $\mathbb{R}^d$ is computed through SVD for each possible edge while in our pruning subroutine, we compute the rank of $\widehat{C}_{i,j} \in \mathbb{R}^{K \times (i-j)}$ and $\widetilde{C}_{i,j} \in \mathbb{R}^{K \times (i-j+1)}$ for each possible edge from node $i$ to node $j$ with $i \geq j + 1$. Concretely, in the pruning step, we regress $\widehat{z}^{(k)}$ against $\widetilde{y}^{(k)}$ and denote the regression coefficient as $\widehat{B}^{(k)} \in \mathbb{R}^{d \times d}$. For any different $1 \leq j \leq i - 1 \leq d$, we construct $\mathbb{R}^K \ni \widehat{B}_{i,j} :=$

---

[1]This part of the proof adapts the proof of Theorem 1 in Jin & Syrgkanis (2024) to our context.

$(\widehat{B}_{i,j}^{(k)}, k \in [K])$ and $\mathbb{R}^{K \times (i-j)} \ni \widehat{C}_{i,j} := (\widehat{B}_{i,l}, l \in \{j+1, \dots, i\})$ and $\widetilde{C}_{i,j} := (\widehat{B}_{i,j}, \widehat{C}_{i,j})$. Next, we compute the rank of $\widehat{C}_{i,j}$ and $\widetilde{C}_{i,j}$ and if and only if $\text{rank}(\widehat{C}_{i,j}) = \text{rank}(\widetilde{C}_{i,j}) - 1$, we conclude that $j \in \text{pa}(i)$. As $\widetilde{C}_{i,j} = \widehat{C}_{i,j+1}$, we actually need to compute only $\widetilde{C}_{i,j}$ for $j \geq 2$. In this regard, we claim that our pruning method is more efficient.

---

**Algorithm 3** Disentanglement

**Input:** estimated latent features $\widetilde{Y}^{(k)}$, noise $\widehat{Z}^{(k)}$ for $k \in [K]$, estimated causal DAG $\widehat{G} = (\widehat{V}, \widehat{E})$ from Algorithm 2
**Output:** Disentangled latent feature $\widehat{y}^{(k)}$ such that $\widehat{y}^{(k)} \sim_{\text{sur}} y^{(k)}$

1: **for all** $k \in \{1, \dots, K\}$ **do**
2:     regress $\widehat{Z}^{(k)}$ on $\widetilde{Y}^{(k)}$ and denote the regression coefficient as $\widehat{B}^{(k)}$
3: **end for**
4: **for all** $i \in \{1, \dots, d\}$ **do**
5:     $\mathcal{V}_i = \text{span}\{\widehat{B}_{i,\cdot}^{(k)} : k \in [K]\}$
6: **end for**
7: **for all** $i \in \{1, \dots, d\}$ **do**
8:     $\breve{B}_{i,\cdot} \leftarrow$ any nonzero vector in $\bigcap_{j \in \overline{\text{ch}}(i)} \mathcal{V}_j$
9: **end for**
10: $\breve{B} \leftarrow (\breve{B}_{i,\cdot} : i \in [d])^\top$
11: **for all** $k \in [K]$ **do**
12:     $\widehat{Y}^{(k)} \leftarrow \widetilde{Y}^{(k)} \breve{B}$
13: **end for**

---

# D   PROOF OF THEOREMS IN SECTION 3

**Proof of Theorem 2**

*Proof.* In the proof, we omit the subscript $i$ in $\alpha_i$, where $i$ denotes the $i$-th iteration. As a preparation for the proof, we denote $\beta := H^\top \alpha$ and $\gamma^{(k)} := \left(\beta^\top (I - W^{(k)\top})^{-1} \Omega^{(k)}\right)^\top$. We also define index sets $I_0 := \{i : \beta_i \neq 0\}$, $I^{(k)} := \{i : \gamma_i^{(k)} \neq 0\}$ and $J^{(k)} := \{i : \gamma_i^{(k)} = 0\}$.

First, we prove that $\#I^{(k)} = 1$. Denote $M^{(k)} := H(I - W^{(k)\top})^{-1} \Omega^{(k)}$ so $\alpha^\top M^{(k)} \equiv \gamma^{(k)\top}$. Then $\forall k \in [K]$, we have $\alpha^\top x^{(k)} = \gamma_{I^{(k)}}^{(k)\top} z_{I^{(k)}}^{(k)}$ and

$$x^{(k)} - \mathbb{E}(x^{(k)} | \alpha^\top x^{(k)}) = M_{\cdot, I^{(k)}}^{(k)} z_{I^{(k)}}^{(k)} + M_{\cdot, J^{(k)}}^{(k)} z_{J^{(k)}}^{(k)} - \mathbb{E}(M_{\cdot, I^{(k)}}^{(k)} z_{I^{(k)}}^{(k)} | \gamma_{I^{(k)}}^{(k)\top} z_{I^{(k)}}^{(k)}) - \mathbb{E}(M_{\cdot, J^{(k)}}^{(k)} z_{J^{(k)}}^{(k)}),$$

where the last marginal mean is due to the independence between $z_{J^{(k)}}^{(k)}$ and $z_{I^{(k)}}^{(k)}$. When $\forall k \in [K]$ $\alpha^\top x^{(k)}$ is independent with $x^{(k)} - \mathbb{E}(x^{(k)} | \alpha^\top x^{(k)})$, by Darmois-Skitovitch theorem, the terms of $z_{I^{(k)}}$ must be zero, implying that $M_{\cdot, I^{(k)}}^{(k)} z_{I^{(k)}}^{(k)} - \mathbb{E}(M_{\cdot, I^{(k)}}^{(k)} z_{I^{(k)}}^{(k)} | \gamma_{I^{(k)}}^{(k)\top} z_{I^{(k)}}^{(k)}) = 0$. In our implementation, $\alpha$ that satisfies the desired conditional independence assumption is a globally optimal solution to the optimization problem (2). Without loss of generality, we assume that $\text{Cov}(z^{(k)}) = I_d \ \forall k \in [K]$, so we have

$$M_{\cdot, I^{(k)}}^{(k)} z_{I^{(k)}}^{(k)} - \mathbb{E}(M_{\cdot, I^{(k)}}^{(k)} z_{I^{(k)}}^{(k)} | \gamma_{I^{(k)}}^{(k)\top} z_{I^{(k)}}^{(k)}) = \left(M_{\cdot, I^{(k)}}^{(k)} - \frac{M_{\cdot, I^{(k)}}^{(k)} \gamma_{I^{(k)}}^{(k)} \gamma_{I^{(k)}}^{(k)\top}}{\gamma_{I^{(k)}}^{(k)\top} \gamma_{I^{(k)}}^{(k)}}\right) z_{I^{(k)}}^{(k)} = 0.$$

As $H$ is of full column rank, we have $M^{(k)}$ and $M_{\cdot, I^{(k)}}^{(k)}$ are also of full column rank and thus $I_{\#I^{(k)}} - \frac{\gamma_{I^{(k)}}^{(k)} \gamma_{I^{(k)}}^{(k)\top}}{\gamma_{I^{(k)}}^{(k)\top} \gamma_{I^{(k)}}^{(k)}} = 0$. As the rank of $\frac{\gamma_{I^{(k)}}^{(k)} \gamma_{I^{(k)}}^{(k)\top}}{\gamma_{I^{(k)}}^{(k)\top} \gamma_{I^{(k)}}^{(k)}}$ is 1, we have $\#I^{(k)} = 1$.

Denote the nonzero index of $\gamma^{(k)}$ as $i^{(k)}$ and thus we have $\beta = U_{i^{(k)}, \cdot}^{(k)\top} \gamma_{i^{(k)}}^{(k)}$. We now claim that $i^{(k)}$ is invariant in $k \in \{1, \cdots, K\}$ and will prove it by contradiction. Assume that on the contrary $\exists k_1 \neq k_2$

such that $i^{(k_1)} \neq i^{(k_2)}$. Since $\gamma_{i^{(k_1)}}^{(k_1)\top} U_{i^{(k_1)},\cdot}^{(k_1)} = \gamma_{i^{(k_2)}}^{(k_2)\top} U_{i^{(k_2)},\cdot}^{(k_2)} = \beta^\top$ and $\forall k$, and by the definition of $U^{(k)}$, diagonal entries of $U^{(k)}$ are all non-zero diagonal elements, we have $\beta_{i^{(k_1)}} \neq 0$ and $\beta_{i^{(k_2)}} \neq 0$. It in turn follows from straightforward algebra in vector-matrix multiplication that $U_{i^{(k_2)},i^{(k_1)}}^{(k_2)} \neq 0$ and $U_{i^{(k_1)},i^{(k_2)}}^{(k_1)} \neq 0$, which further implies that $W_{i^{(k_1)},i^{(k_2)}}^{(k_2)} \neq 0$ and $W_{i^{(k_2)},i^{(k_1)}}^{(k_1)} \neq 0$. However, since $W^{(k)}$ encodes the adjacency matrix of the DAG $\mathcal{G}$ corresponding to the causal model, only one of $W_{i,j}^{(k)}$ and $W_{j,i}^{(k)}$ can be non-zero for every $i \neq j$. We have a contradiction.

Now we can identify all $i^{(k)}$'s by a single node index $i$. Therefore

$$\dim \operatorname{span}\{U_{i,\cdot}^{(k)}, k \in [K]\} = \dim \operatorname{span}\left\{\frac{\beta}{\gamma_i^{(k)}}, k \in [K]\right\} = 1.$$

Finally, by Assumption 2, node $i$ is a root node in $\mathcal{G}$. $\qquad\square$

**Proof of Theorem 3**

*Proof.* As our goal is to learn the latent features and the causal DAG up to permutation and scale, we can assume that $\{1, \cdots, d\}$ is a valid topological ordering of the causal DAG $\mathcal{G}$.

In total, there are $d$ iterations in Algorithm 1. In the first iteration, by Theorem 2, we recover one component of $z^{(k)}$ denoted as $z_1^{(k)}$. Furthermore, $\beta$ corresponding to $H^\top \alpha$ shall be $(\beta_1, 0, \ldots, 0)$. We then eliminate the influence of $z_1^{(k)}$ on $x^{(k)}$ by the orthogonal projection $x^{(k)} - \mathbb{E}(x^{(k)}|z_1^{(k)})$, denoted as $\operatorname{proj}_1^\perp x^{(k)}$. Graphically, this orthogonal projection removes node 1 and associated edges from $\mathcal{G}$ and we denote the new DAG as $\mathcal{G}'$. Algebraically, $\operatorname{proj}_1^\perp x^{(k)} = H\operatorname{proj}_1^\perp y^{(k)} = H(I - W^{(k)\top})^{-1}\Omega^{(k)}\operatorname{proj}_1^\perp z^{(k)}$, meaning that $\operatorname{proj}_1^\perp x^{(k)}$ is the observed data obtained from the corresponding latent feature $\operatorname{proj}_1^\perp y^{(k)}$ in the new causal DAG $\mathcal{G}'$. Therefore, we can repeat the procedure until we estimate all components of $z^{(k)}$ and entangled latent feature $y^{(k)}$ and causal graph $\mathcal{G}$.

Note that in the $i$-th iteration, $\beta^\top \operatorname{proj}_i^\perp y^{(k)} = z_i^{(k)}$ and thus $\beta_i \neq 0$ and $\beta_{(i+1):d} = 0$. Therefore, the recovered $\widetilde{y}_i^{(k)}$ is a linear combination of $\{y_1^{(k)}, \ldots, y_i^{(k)}\}$, which in turn implies that there exists a lower triangular matrix $B$ such that $\widetilde{y}^{(k)} = By^{(k)}$. Since all the $\widehat{\alpha}_i$, $i \in [d]$, are estimated through an ICA algorithm, by Theorem 11 in Reyhani et al. (2012), we identify all components of $z^{(k)}$ up to permutation and scale. $\qquad\square$

**Proof of Theorem 4**

*Proof.* Suppose that $\widehat{z}^{(k)}$ and $\widetilde{y}^{(k)}$ are perfectly-solved output from subroutine 1. Then $\widehat{z}^{(k)} = z^{(k)}$ and $\widetilde{y}^{(k)} = By^{(k)}$ where $B$ is a lower triangular matrix. Therefore, we have $\widehat{z}^{(k)} = \Omega^{(k)-1}(I - W^{(k)})^\top B^{-1}\widetilde{y}^{(k)}$ and thus $\widehat{B}^{(k)} = \Omega^{(k)-1}(I - W^{(k)})^\top B^{-1}$, following that

$$\widehat{B}_{i,j}^{(k)} = ((\Omega^{(k)})^{-1})_{i,i}(B^{-1})_{\cdot,j}^\top (e_i - W_{\cdot,i}^{(k)}). \tag{4}$$

For any $i, j \in [d]$, denote the vectors $(\widehat{B}_{i,j}^{(1)}, \widehat{B}_{i,j}^{(2)}, \ldots, \widehat{B}_{i,j}^{(K)})$, $(W_{i,j}^{(1)}, W_{i,j}^{(2)}, \ldots, W_{i,j}^{(K)})$, $((\Omega_{i,i}^{(1)})^{-1}, (\Omega_{i,i}^{(2)})^{-1}, \ldots, (\Omega_{i,i}^{(K)})^{-1})$, and $(W_{i,j}^{(1)}(\Omega_{i,i}^{(1)})^{-1}, W_{i,j}^{(2)}(\Omega_{i,i}^{(2)})^{-1}, \ldots, W_{i,j}^{(K)}(\Omega_{i,i}^{(K)})^{-1})$ as $\widehat{B}_{i,j}$, $W_{i,j}$, $\Omega_i^\dagger$, and $W_{i,j}^\Omega$. Since $B$ is a lower triangular matrix, we have $(B^{-1})_{1:j-1,j} = 0$. As we suppose the $\widehat{z}^{(k)}$ and $\widetilde{y}^{(k)}$ are estimated in topological order, $W^{(k)}$ is an upper triangular matrix and thus $(e_i - W_{\cdot,i}^{(k)})_{i+1:d} = 0$. Together we have that $\widehat{B}_{i,j}^{(k)} = \Omega_{i,i}^{(k)-1}\sum_{j'=j}^{i}(B^{-1})_{l,j}(e_i - W_{\cdot,i}^{(k)})_{j'}$.

Therefore, $\widehat{B}_{i,j}$ is a linear combination of $(W_{i,j'}^\Omega, j' \in \{j, \ldots, i\})$ and $\Omega_i^\dagger$. Similarly, we obtain that for any $l \in \{j+1, \ldots, i\}$, $\widehat{B}_{i,l}$ is a linear combination of $(W_{i,l'}^\Omega, l' \in \{l, \ldots, i\})$ and $\Omega_i^\dagger$. Therefore, for any different $j, i \in [d]$ such $j \notin \operatorname{pa}(i)$, we have $W_{j,i} = 0$. As the diagonal entries of $B$ is nonzero, $\widehat{B}_{i,j}$ is a linear combination of vectors $\widehat{B}_{i,l}, \forall l \in \{j+1, \ldots, i\}$, implying that $\operatorname{rank}(\widehat{C}_{i,j}) = \operatorname{rank}(\widetilde{C}_{i,j})$.

When $j \in \mathrm{pa}(i)$, $W_{i,j} \neq 0$, and with Assumption 2, we could obtain $\mathrm{rank}(\widehat{C}_{i,j}) = \mathrm{rank}(\widetilde{C}_{i,j}) - 1$. Therefore, we can conclude that $j \in \mathrm{pa}(i)$ if and only if

$$\mathrm{rank}(\widehat{C}_{i,j}) = \mathrm{rank}(\widetilde{C}_{i,j}) - 1.$$

$\square$

**Proof of Theorem 5**

*Proof.* Suppose that $\widetilde{y}^{(k)}$ and $\widehat{G}$ are perfectly solved in the previous subroutine 1 and 2 with the same topological ordering as the ground truth (without loss of generality), meaning that $\widetilde{y}^{(k)} = By^{(k)}$ where $B$ is an lower triangular matrix and $\widehat{G} = G$. Thus, we have $\widehat{z}^{(k)} = (\Omega^{(k)})^{-1}(I - W^{(k)})^{\top}B^{-1}\widetilde{y}^{(k)}$. We regress $\widehat{z}^{(k)}$ on $\widetilde{y}^{(k)}$ and let $\widehat{B}^{(k)}$ denote the regression coefficient. Consequently, the $i$th row vector of $\widehat{B}^{(k)}$ is given by $B^{-1}(e_i - W^{(k)}_{\cdot,i})(\Omega^{(k)-1})_{i,i}$. We obtain that $\widehat{B}^{(k)}_{i,\cdot} \in \mathcal{V}_i := \mathrm{span}\{B_{i,\cdot} : i \in \overline{\mathrm{pa}}(i)\}$. Together with Assumption 2, $\dim(\mathrm{span}\{(B^{-1})_{i,\cdot} : i \in \overline{\mathrm{pa}}(i)\}) \leq |\overline{\mathrm{pa}}(i)| = \mathcal{V}_i$, which implies that $\mathcal{V}_i = \mathrm{span}\{(B^{-1})_{i,\cdot} : i \in \overline{\mathrm{pa}}(i)\}$. Recall that $\mathrm{sur}(i) \equiv \mathrm{pa}(i) \cap (\bigcap_{j \in \mathrm{ch}(i)} \mathrm{pa}(j))$. Therefore, $\bigcap_{j \in \overline{\mathrm{ch}}(i)} \mathcal{V}_j = \mathrm{span}\{B_{i,\cdot} : i \in \overline{\mathrm{sur}}(i)\}$. As $y^{(k)} = B^{-1}\widetilde{y}^{(k)}$, we denote the estimated latent features as $\widehat{y}^{(k)}$ and it reads

$$\widehat{y}^{(k)}_i := \widecheck{B}^{\top}_{i,\cdot}\widetilde{y}^{(k)} = \sum_{j \in \overline{\mathrm{sur}}(i)} \widecheck{B}_{i,j}y^{(k)}_j,$$

where $\widecheck{B}_{i,\cdot}$ is any nonzero vector in $\bigcap_{j \in \overline{\mathrm{ch}}(i)} \mathcal{V}_j$. $\square$

## D.1 Convergence Analysis of Algorithm 1

Denote the topological ordering obtained by subroutine 1 as $\widehat{\pi}$. We first present the convergence analysis of $\widehat{\pi}$.

**Lemma 3.** *Without loss of generality, we assume that for any node $i < j$, node $i$ is not a descendant of $j$. Denote the topological ordering output by subroutine 1 as $\widehat{\pi}$ from $X^{(k)} \in \mathbb{R}^{n \times p}$, $\forall k \in [K]$ and the set of all possible ground truths as $\Pi$. We have*

$$\lim_{n \to \infty} \mathbb{P}(\widehat{\pi} \in \Pi) = 1. \tag{5}$$

*Proof.* Since $\widehat{\pi}$ is obtained by $d$ steps sequentially, we only need to prove that the probability of the estimated latent variable in the $i$-th step $\widehat{y}^{(k)}_i$ corresponds to a root node of the subgraph with the first $i-1$ nodes removed in $G$ tends to 1 as sample size $n$ tends to infinity. For the first step, denote all $K \cdot d$ many possible candidates from the ICA algorithm as $\widehat{\alpha}_{1,j}$, $j \in [K \cdot d]$. By Theorem 2, we only need to prove that the mutual information estimator tends to the ground truth with probability converging to 1 as the sample size $n \to \infty$. As we only need to find $\alpha$ such that $\alpha^{\top}x^{(k)}$ is independent with $x^{(k)} - \mathbb{E}(x^{(k)}|\alpha^{\top}x^{(k)})$, in our algorithm, we replace mutual information with HSIC estimator (Gretton et al., 2005), which is an independence criterion, satisfying that if and only if the two random variables are independent, the estimator would be 0. We denote the estimator of HSIC and the true value of HSIC as HSIC and hsic, respectively.

Up to this point, we are left to show that $\forall \varepsilon > 0$,

$$\sum_{k=1}^{K}\sum_{i=1}^{d} \mathrm{hsic}(X^{(k)}_{\cdot,i}\widehat{\alpha}, X^{(k)}_{\cdot,i} - \widehat{\mathbb{E}}(X^{(k)}_{\cdot,i}|X^{(k)}_{\cdot,i}\widehat{\alpha})) \to_P \sum_{k=1}^{K}\sum_{i=1}^{d} \mathrm{HSIC}(\alpha^{\top}x^{(k)}_i, x^{(k)}_i - \mathbb{E}(x^{(k)}_i|\alpha^{\top}x^{(k)}_i)), \tag{6}$$

where $\widehat{\mathbb{E}}$ denotes the estimated version of mean or conditional mean operator.

As all $\alpha$ candidates in (2) are from the row vectors of the unmixing matrix of ICA, by the consistency of the estimated unmixing matrix (Reyhani et al., 2012), we have that $\widehat{\alpha} \to_P \alpha$. Without loss of generality, we assume that $\mathbb{E}(x^{(k)}) = 0$, $\forall k \in [K]$. If not, we could replace $x^{(k)}$ by $x^{(k)} - \mathbb{E}(x^{(k)})$ during implementation. Then we have $\frac{1}{n}(X^{(k)}\widehat{\alpha})^{\top}X^{(k)}\widehat{\alpha} \to_P \mathrm{Var}(\alpha^{\top}x^{(k)})$ and $\frac{1}{n}(X^{(k)}\widehat{\alpha})^{\top}X^{(k)}_i \to_P$

$\mathrm{Cov}(\alpha^\top x^{(k)}, x_i^{(k)}) \ \forall i \in [d], \ k \in [K]$. Therefore, we have $((X^{(k)}\widehat{\alpha})^\top X^{(k)}\widehat{\alpha})^{-1}(X^{(k)}\widehat{\alpha})^\top X_i^{(k)} \to_P$

$\frac{\mathrm{Cov}(\alpha^\top x^{(k)}, \ x_i^{(k)})}{\mathrm{Var}(\alpha^\top x^{(k)})} \ \forall i \in [d], \ k \in [K]$. By the definition of empirical HSIC in (Gretton et al., 2005), we

could know that $\mathrm{hsic}(X^{(k)}\widehat{\alpha}, X_{\cdot,i}^{(k)} - \mathbb{E}(X_{\cdot,i}^{(k)} | X^{(k)}\widehat{\alpha})) - \mathrm{hsic}(X^{(k)}\widehat{\alpha}, X_{\cdot,i}^{(k)} - \frac{\mathrm{Cov}(\alpha^\top x^{(k)}, \ x_i^{(k)})}{\mathrm{Var}(\alpha^\top x^{(k)})} X^{(k)}\widehat{\alpha}) =$

$\frac{1}{(n-1)^2}\mathrm{tr}KH(L'-L)H$, where $H, K, L, L' \in \mathbb{R}^{n \times n}$ and are defined as $\forall l_1, l_2 \in [n]$

$$K_{l_1,l_2} := k(X_{l_1,\cdot}^{(k)}\widehat{\alpha}, X_{l_2,\cdot}^{(k)}\widehat{\alpha}),$$

$$L_{l_1,l_2} := l(X_{l_1,i}^{(k)} - \frac{\mathrm{Cov}(\alpha^\top x^{(k)}, \ x_i^{(k)})}{\mathrm{Var}(\alpha^\top x^{(k)})}X_{l_1,\cdot}^{(k)}\widehat{\alpha}, X_{l_2,i}^{(k)} - \frac{\mathrm{Cov}(\alpha^\top x^{(k)}, \ x_i^{(k)})}{\mathrm{Var}(\alpha^\top x^{(k)})}X_{l_2,\cdot}^{(k)}\widehat{\alpha}),$$

$$L' := l(X_{l_1,i}^{(k)} - \mathbb{E}(X_{l_1,i}^{(k)} | X_{l_1,\cdot}^{(k)}\widehat{\alpha}), X_{l_2,i}^{(k)} - \mathbb{E}(X_{l_2,i}^{(k)} | X_{l_2,\cdot}^{(k)}\widehat{\alpha})),$$

and

$$H_{i,j} := \delta_{i,j} - \frac{1}{n},$$

with $l(\cdot, \cdot)$ and $k(\cdot, \cdot)$ kernel function. In our implementation, we leverage RBF kernel, which is a bounded continuous function, implying that $l'_{i,j} \to_P l_{i,j}$. Thus we can obtain that $\mathrm{hsic}(X^{(k)}\widehat{\alpha}, X_{\cdot,i}^{(k)} - \mathbb{E}(X_{\cdot,i}^{(k)} | X^{(k)}\widehat{\alpha})) \to_P \mathrm{hsic}(X^{(k)}\widehat{\alpha}, X_{\cdot,i}^{(k)} - \frac{\mathrm{Cov}(\alpha^\top x^{(k)}, \ x_i^{(k)})}{\mathrm{Var}(\alpha^\top x^{(k)})}X^{(k)}\widehat{\alpha})$. By Theorem 3 in Gretton et al. (2005), we have

$$\mathrm{hsic}(X^{(k)}\widehat{\alpha}, X_{\cdot,i}^{(k)} - \mathbb{E}(X_{\cdot,i}^{(k)} | X^{(k)}\widehat{\alpha})) \to_P \mathrm{HSIC}(\alpha^\top x^{(k)}, x_i^{(k)} - \mathbb{E}(x_i^{(k)} | \alpha^\top x^{(k)})).$$

Therefore, the probability of estimated $\widetilde{y}_1^{(k)}$ and $\widehat{z}_1^{(k)}$ correspond to a root node tends to 1 as the sample size tends to infinity. With a similar proof, it can be shown that $\lim_{n\to\infty} \mathbb{P}(\widehat{\pi} \in \Pi) = 1$. $\square$

**Theorem 6.** *For the estimated* $(\widehat{y}^{(k)}, \widehat{\mathcal{G}})$ *from Algorithm* 1*, we have*

$$\lim_{n\to\infty} \mathbb{P}((\widehat{y}^{(k)}, \widehat{\mathcal{G}}) \sim_{\mathrm{sur}} (y^{(k)}, \mathcal{G})) = 1, \ \forall k \in [K]. \tag{7}$$

*Proof.* Since the estimated $\widehat{\alpha} \to_P \alpha$, we have $\lim_{n\to\infty} \mathbb{P}(\widehat{z}^{(k)} \sim_P z^{(k)}) = 1$ and $\lim_{n\to\infty} \mathbb{P}(\widetilde{y}^{(k)} \sim_\triangle y^{(k)}) = 1$, implying that the estimated $\widehat{B}^{(k)}$ in Algorithm 2 and 3 is converging in probability. Together with the results in Lemma 3, we have that $\lim_{n\to\infty} \mathbb{P}((\widehat{y}^{(k)}, \widehat{\mathcal{G}}) \sim_{\mathrm{sur}} (y^{(k)}, \mathcal{G})) = 1, \ \forall k \in [K]$. $\square$

### D.2 COMPUTATIONAL COMPLEXITY OF CREATOR

In subroutine 1, the computational cost is mainly due to ICA and the step of computing independence criterion (HSIC in the current version), resulting in an overall computational complexity of $O(pn^3d)$. Subroutine 2 involves regression and rank estimation. Specifically, Singular Value Decomposition (SVD) is used to determine the rank by counting the number of positive singular values, leading to complexity $O(n^2d^3)$. In subroutine 3, we employ a method analogous to the one outlined in Section B.2 of Jin & Syrgkanis (2024). This involves computing the orthogonal projection matrix of $\mathcal{V}_j$, denoted as $Q_j$, and extracting the singular vector associated with the least singular value of $\sum_{j\in\mathrm{ch}(i)} Q_j^\top Q_j$. Thus, the computational cost of subroutine 3 is also $O(n^2d^3)$, resulting in a total computational cost $O(pn^3d + n^2d^3)$.

## E SUPPLEMENTARY INFORMATION ON NUMERICAL EXPERIMENTS

### E.1 METRICS

We use structural Hamming distance (SHD) for causal DAGs. SHD counts the number of missing, falsely detected or reversed edges. For latent features, we design a metric called LocR$^2$ closely related to $R^2$:

$$\mathrm{LocR}^2 := \max_P \frac{1}{dK} \sum_{k=1}^{K} \sum_{i=1}^{d} \mathrm{LocR}^2_{i,k}, \quad \mathrm{LocR}^2_{i,k} := 1 - \frac{\widehat{\mathbb{E}}(\check{y}_i^{(k)} - \mathrm{proj}^\perp_{\mathrm{span}(y_j^{(k)}:j\in\mathrm{sur}(j))}(\check{y}_i^{(k)}))^2}{\widehat{\mathrm{Var}}(\check{y}_i^{(k)})},$$

where $\breve{y}^{(k)} := P\widehat{y}^{(k)}$ for some permutation matrix $P$, and $\widehat{\mathbb{E}}$ and $\widehat{\mathrm{Var}}$ denote, respectively, the sample mean and sample variance. $\mathrm{LocR}_i^2$ measures the linear correlation between $\breve{y}_i^{(k)}$ and $(y_j^{(k)}, j \in \mathrm{sur}(i))$. When $\mathrm{LocR}_i^2$ is close to 1, $\breve{y}_i^{(k)}$ is close to $\mathrm{span}\{y_j^{(k)} : j \in \mathrm{sur}(i)\}$; when $\mathrm{LocR}^2$ is 1, $\widehat{y}^{(k)} \sim_{\mathrm{sur}} y^{(k)}$.

For SHD, lower is better while for $\mathrm{LocR}^2$, higher is better.

## E.2 OTHER RESULTS IN SECTION 4.1

In this section, we first describe the simulation settings in more details. The weighted matrices $W^{(k)}$ are generated in two steps. First, we generate a directed acylic graph based on the Erdős-Rényi random graph model (Erdős & Rényi, 1959) and obtain its adjacency matrix. Then we generate a random matrix as the weight matrix. We generate the weight matrix randomly from several non-Gaussian distributions, listed in Table 2. For each weight matrix, we first randomly select a distribution and then generate the corresponded random matrix. Each entry of the weight matrix is independently drawn. After we generate these two matrices, we obtain $W^{(k)}$ by multiplying the corresponding entries of the two matrices.

Table 2: Distributions and Their Parameters

| Distribution | Parameters |
|---|---|
| Laplace | Location 0, Scale 1 |
| Exponential | Rate 1 |
| Uniform | Lower bound 0, Upper bound 1 |
| Gumbel | Location 0, Scale 1 |
| Beta | Shape 0.5, Shape 0.5 |
| Gamma-1 (Gamma with shape=1) | Shape 1, Scale 1 (or Rate $\beta = 1/\theta$) |
| Chi-squared-1 ($\chi_1^2$) | Degrees of freedom 1 |
| Chi-squared-3 ($\chi_3^2$) | Degrees of freedom 3 |
| Gamma-3 (Gamma with shape=3) | Shape $k = 3$, Scale 1 |

We then present Figure 3, which is still on the synthetic experiments conducted in Section 4.1 of the main text, but with $K = 2d$. The overall pattern is quite similar to the results in the main text so we do not further expound upon it.

## E.3 THE IMPACT OF INFERRING TOPOLOGICAL ORDERING

In this section, we design ablation experiments to compare the performance of CREATOR across various settings in which we expect that the accuracies should differ in inferring topological ordering. Specifically, the results show that poor accuracy in inferring topological ordering could lead to poor latent causal feature recovery. Here, we choose the topological divergence in Rolland et al. (2022) as the metric for topological ordering, defined as $D_{\mathrm{top}}(\pi, W) := \sum_{i=1}^{d} \sum_{j:\pi(i)>\pi(j)} W_{ij}$ where $W$ is the adjacency matrix of $\mathcal{G}$ with binary value. From (1), we conclude that for any two nodes $i, j \in [d]$, when $w_{i,j}^{(k)}$ is close to 0, the identification of causal order would be harder than the situation where $w_{i,j}^{(k)}$ is positive. The reason might be weak causal effect is similar to non-causal effect and could confuse the algorithm. With this intuition, we generate data like the general case in last subsection but choose smaller standard deviation for the weights of the causal DAG $w^{(k)}$ by multiplying the generated data with $\sigma \in \{0.005, 0.007, 0.01, 0.03, 0.05, 0.07, 0.1, 0.3, 0.5\}$. We repeat each simulation setting 50 times and report the average values of $\mathrm{LocR}^2$ and topological divergence. The results for $K = 2d$ and $K = d$ are presented respectively in Figure 4. From the results we conclude that more accurate topological ordering inference leads to more accurate recovery of

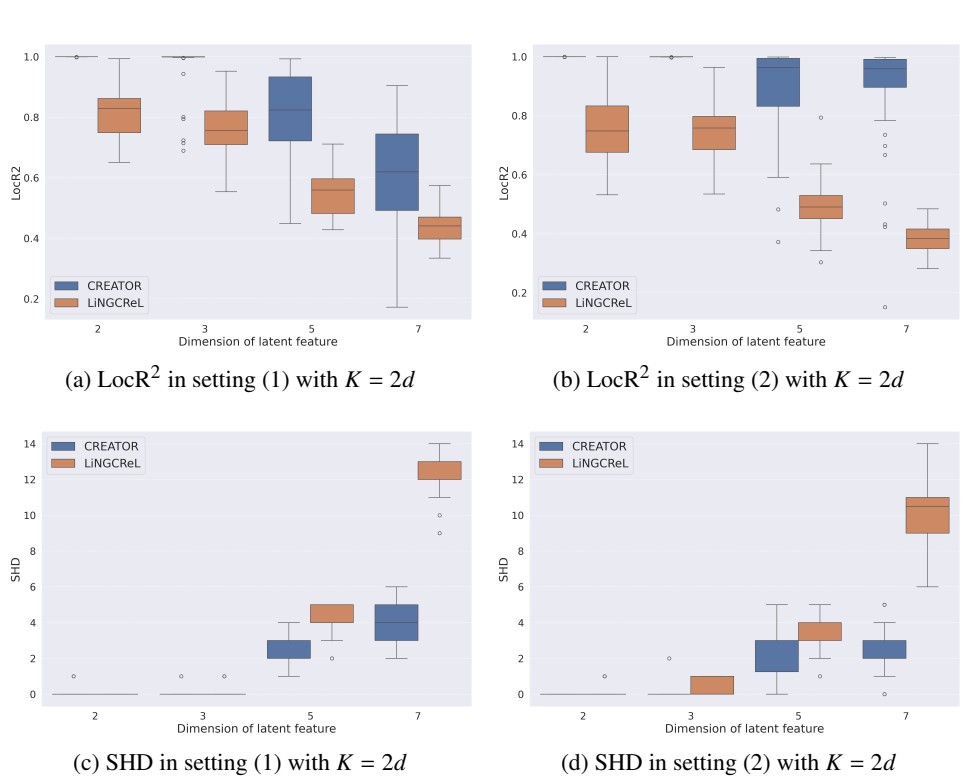

(a) LocR$^2$ in setting (1) with $K = 2d$

(b) LocR$^2$ in setting (2) with $K = 2d$

(c) SHD in setting (1) with $K = 2d$

(d) SHD in setting (2) with $K = 2d$

Figure 3: LocR$^2$ and SHD metric for different data generation setup. Figures 2a and 2c compare the performance of latent feature and causal DAG identification in setting (1). Figures 2b and 2d compare the performance in setting (2).

latent causal features in most cases, suggesting the value of first inferring topological ordering in CREATOR.

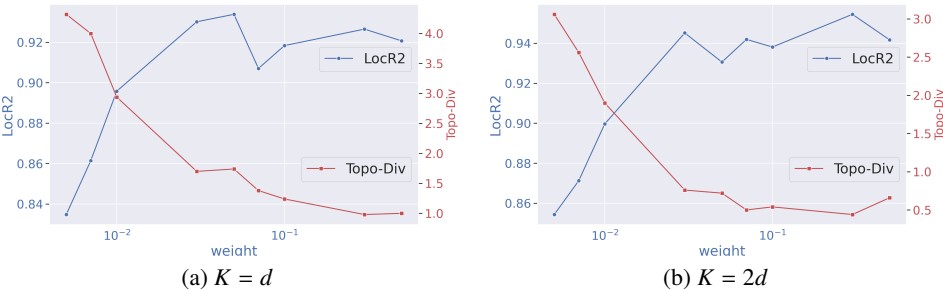

(a) $K = d$          (b) $K = 2d$

Figure 4: The impact of topological ordering inference on the performance of CREATOR.

### E.4 IMPLEMENTATION DETAILS AND OTHER RESULTS IN SECTION 4.2

The goal of the real data analysis conducted in Section 4.2 is to illustrate how even linear CRL methods (e.g. CREATOR and/or LiNGCReL) can be useful to help us unpack the black-box of large language models (LLMs).

To this end, we employ GPT-4 and DeepSeek to generate three types of stories (so $K = 3$), with sufficient diversity in their styles, including news ($k = 1$), fairy tales ($k = 2$), and plain texts ($k = 3$). For each style, we generate $n = 900$ different stories with different BG, CD and ED with GPT-4 and DeepSeek using the prompt: "Generate {$n$} {style} English stories, each containing background, condition, and ending, with each story limited to 100 words or less. Output the content, the keywords for background, the keywords for condition, and the keywords for ending, with keywords restricted to 2-3-word strings. Format your output for easy copy-pasting into a JSON file, ensuring it only includes the content, background keywords, condition keywords, and ending keywords." in which {$n$} and {style} are the number and style of stories to be generated.

We input the generated stories to open source LLMs that we mentioned in Table 1 and extract the last hidden states from the corresponding LLMs. As these hidden states can be extremely high-dimensional (mostly around $2048 \times 25$ for the models used in this paper), we reduce the data dimension in two steps. First, we multiply these them by a matrix i.i.d. drawn from standard Gaussian distribution with column number $p = 2$ to reduce the dimension to 2048 and flatten the last two dimensions into one. Then they are in turn multiplied by a random matrix i.i.d. drawn from standard Gaussian distribution with column number $p = 30$. The two dimensionality reduction steps borrow idea from the sketching randomized algorithm literature (Woodruff, 2014; Larsen & Nelson, 2017). We use the data after dimension reduction as the observed data matrix $X^{(k)} \in \mathbb{R}^{n \times p}$. We then obtain the estimated latent causal features $\widehat{y}^{(k)}$ and the DAG $\widehat{\mathcal{G}}$ using either CREATOR or LiNGCReL.

Next we prepare the proxy labels using the generated keywords of each story from the LLM output. We also input the generated BG, CD, and ED to the same LLMs and extract the last hidden states.

To evaluate the performance of our algorithm, we first need to find a particular permutation because the returned latent features are only up to $\sim_{\text{sur}}$ equivalence. We first identify the BG feature as it is not entangled with other features. We use the extracted hidden states of BG as input of our neural network and one of the estimated features as label. Then we select the estimated feature with the least average test loss as the BG feature. Then we use the extracted hidden states of BG and CD as input and one of the other two estimated features as label. Similarly, we select the the estimated feature with the least average test loss as the CD feature. The last remaining feature is then automatically selected as the ED feature. The architecture of the neural network is designed as follows. We first permute the last two dimensions of the input features and the number of the three dimensions are respectively batch size, hidden dimension and sequence length. The first part consists of a convolution layer, followed by a batch norm layer, and a ReLU activation function. Then we flatten the last two coordinates of the output from the previous part, which are then transferred as the input to the second part, which

consists of a linear layer, followed by a ReLU activation function, a linear layer again and a dropout operation. The detailed architecture is shown in Table 3 and Table 4. All the layers used in the neural network is from PyTorch (Paszke et al., 2017).

Table 3: Architecture of the convolution module

| Layer | Parameters |
|---|---|
| torch.nn.Conv1d | in_channels = sequence_length, out_channels = 8, kernel_size = 1 |
| torch.nn.BatchNorm1d | num_features = 8 |
| torch.nn.ReLU | - |

Table 4: Architecture of the fully connected module

| Layer | Parameters |
|---|---|
| torch.nn.Linear | in_features = 8 × hidden_dimension, out_features = 8 |
| torch.nn.ReLU | - |
| torch.nn.Linear | in_features = 8, out_features = 1 |
| torch.nn.Dropout | p = 0.5 |

### E.5    NUMERICAL EVALUATIONS FOR CREATOR WHEN THE NOISE DISTRIBUTION IS CLOSER TO AND EXACTLY GAUSSIAN

As Gaussian noise is also very common in real world scenarios, we test our algorithm on data generated with noise variable $z_i^{(k)} = \sigma_i^{(k)} \frac{\epsilon_i^{(k)}}{\sqrt{\mathrm{Var}(\epsilon_i^{(k)})}}$ where for any $i \in [d]$ and $k \in [K]$, $\epsilon_i^{(k)}$ is drawn from general normal distribution with probability density function $p(\epsilon) = \frac{\beta}{\Gamma(1/\beta)} e^{-|\epsilon|^\beta}$. For $\beta = 2$, the noise variable is Gaussian distribution. In our simulation, we set $\beta = 2, 2.1, 2.5$ to compare the performance and show the results in Figure 5. We can see that the performance does not degrade much for Gaussian situation.

## F    POTENTIAL DIRECTION OF GENERALIZING CREATOR TO NONLINEAR MODELS

In our algorithm, an important part of the topological ordering subroutine is to find an $\alpha \in \mathbb{R}^p$ that $\alpha^\top x^{(k)}$ is a exogenous noise variable component and the finding such $\alpha$ rely on the result of Darmois-Skitovitch theorem which states that if two linear forms of independent non-Gaussian random variables are independent, then all the variables with non-zero coefficients in both forms must be normally distributed. Therefore, extension to nonlinear structural causal models is difficult.

One viable approach involves considering a nonlinear additive noise structural causal model (SCM) with a linear mixing function. Under this setup, the latent variable corresponding to the root node can be linearly mapped from the observed data. Following a similar procedure, we could sequentially remove the causal influence of the root node variable on other components of $y^{(k)}$ and iteratively repeat this process.

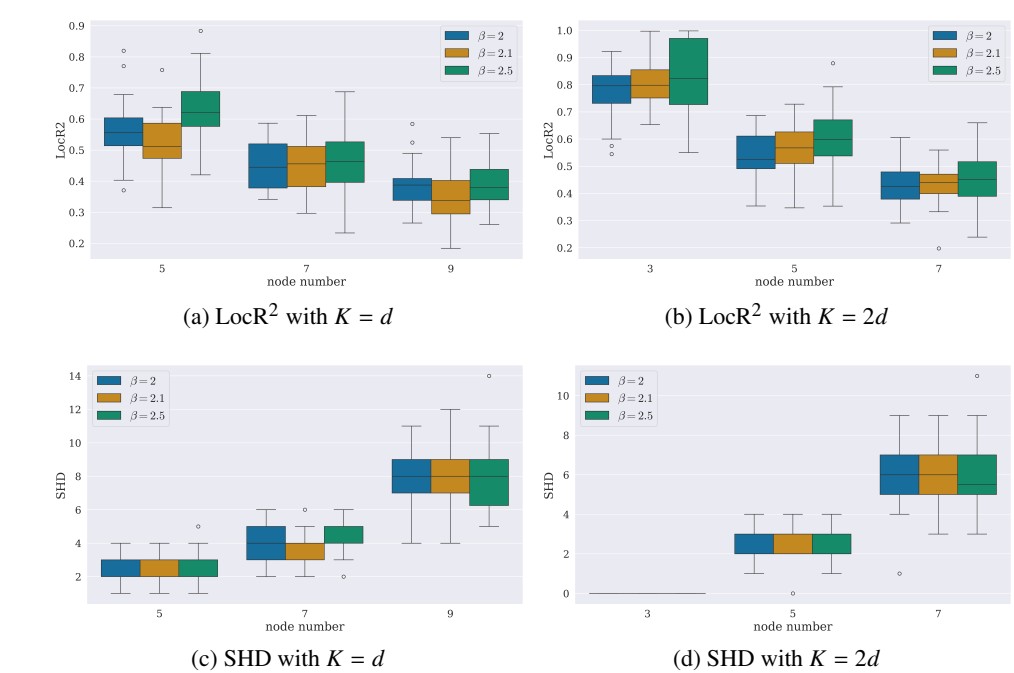

Figure 5: LocR$^2$ and SHD metric for noise variables with Gaussian and non-Gaussian with $K = d$ and $K = 2d$.

## G ILLUSTRATING EXAMPLES FOR FREQUENTLY USED NOTATIONS AND ALGORITHM

### G.1 AN EXAMPLE FOR UNDERSTANDING $\sim_\pi$, $\sim_\triangle$ AND $\sim_{\text{sur}}$

Consider a toy model such that for $\forall k \in [3]$, $x^{(k)} = Hy^{(k)}$, $y^{(k)} = w^{(k)\top}y^{(k)} + \Omega^{(k)}z^{(k)}$ where $H = I$, $w^{(k)}$ is weighted adjacency matrix of $\mathcal{G}$ with two edges including $1 \to 2$ and $2 \to 3$, where $1, 2, 3$ denotes the node indices. When $\widehat{y}^{(k)} \sim_\pi y^{(k)}$ for $k \in [3]$, there exist a permutation $\pi$ and three non-zero number $\lambda_1$, $\lambda_2$ and $\lambda_3$ such that $\widehat{y}_i^{(k)} = \lambda_i y_{\pi(i)}^{(k)}$ for any $i \in [3]$ for all $k \in [3]$.

When $\widehat{y}^{(k)} \sim_\triangle y^{(k)}$ for $k \in [3]$, there exist a lower triangular matrix $B$ and a permutation $\pi$ such that $\widehat{y}_1^{(k)} = B_{1,1}y_{\pi(1)}^{(k)}$, $\widehat{y}_2^{(k)} = B_{2,1}y_{\pi(1)}^{(k)} + B_{2,2}y_{\pi(2)}^{(k)}$, and $\widehat{y}_3^{(k)} = B_{3,1}y_{\pi(1)}^{(k)} + B_{3,2}y_{\pi(2)}^{(k)} + B_{3,3}y_{\pi(3)}^{(k)}$ for all $k \in [3]$. According to the definition of surrounding set, we have $\overline{\text{sur}}_{\mathcal{G}}(1) = \{1\}$, $\overline{\text{sur}}_{\mathcal{G}}(2) = \{2\}$ and $\overline{\text{sur}}_{\mathcal{G}}(3) = \{2, 3\}$. When $\widehat{y}^{(k)} \sim_{\text{sur}} y^{(k)}$ for $k \in [3]$, there exist a lower triangular matrix $B$ and a permutation $\pi$ such that $\widehat{y}_1^{(k)} = B_{1,1}y_{\pi(1)}^{(k)}$, $\widehat{y}_2^{(k)} = B_{2,2}y_{\pi(2)}^{(k)}$, and $\widehat{y}_3^{(k)} = B_{3,2}y_{\pi(2)}^{(k)} + B_{3,3}y_{\pi(3)}^{(k)}$.

### G.2 AN EXAMPLE FOR THE CORE MECHANISM OF THE ALGORITHM

To better explain the intuition of our algorithm, in this section, we present an illustrative example. Consider a toy model in which $K = 3$ and $\forall k \in [3]$, $x^{(k)} = y^{(k)}$, $y^{(k)} = w^{(k)\top}y^{(k)} + z^{(k)}$. Here, we take both $H$ and $\Omega^{(k)}$ to be identities for simplicity, and $w^{(k)}$ is the weighted adjacency matrix of $\mathcal{G}$ representing the causal graph $1 \to 2$ and $2 \to 3$, where $1, 2, 3$ are vertex indices.

In the first subroutine, we run ICA on $x^{(k)}$ for $k \in [3]$ and thus obtain the factorization $x^{(k)} = \widehat{A}^{(k)}\widehat{z}^{\dagger(k)}$, where $\widehat{A}^{(k)}$ is a $3 \times 3$ matrix and $\widehat{z}^{(k)}$ are independent components computed by ICA. As $H$ and $\mathcal{G}$ are invariant across environments, we can find a vector $\alpha_0$ such that $\alpha_0^\top x^{(k)} \propto_k y_r^{(k)}$ where $y_r^{(k)}$ corresponds to some root node, influenced only by the exogeneous noise $z_r^{(k)}$. Therefore, such $\alpha_0$ has to be parallel to one of the row vectors of all three unmixing matrices $\widehat{A}^{(k)}$ for $k \in [3]$. A formal version of this claim can be found in Theorem 2.

Next, we choose one row of all the rows in $\widehat{A}^{(k)}$ across $k \in [3]$, denoted by $\widehat{\alpha}_1$, such that for all $k \in [3]$ and $j \in [3]$, $\widehat{z}_1^{(k)} := \widehat{\alpha}_1^\top x^{(k)}$ is independent of $r_j^{(k)}$, where $r_j^{(k)}$ is the residual of projecting $y_j^{(k)}$ onto $\widehat{z}_1^{(k)}$. $\widehat{\alpha}_1$ is then parallel to $\alpha_0$ because $r_j^{(k)}$ is independent of $\widehat{z}_1^{(k)}$ if and only if $\widehat{z}_1^{(k)}$ equals to a component of $z^{(k)}$ for all $k \in [3]$ up to scale by non-Gaussianity using the Darmois-Skitovitch theorem [4,5]. Furthermore, because the independence condition must be satisfied for all $k \in [3]$, $\widehat{z}_1^{(k)}$ must also be equal to the root node in $y^{(k)}$ up to scale, by the non-degeneracy of $W^{(k)}$. Since $\widehat{z}_1^{(k)}$ corresponds to the root node, $\widetilde{y}_1^{(k)} := \widehat{z}_1^{(k)}$ can serve as an estimator of $y_1^{(k)}$.

Next, given that we have identified the root node in the original causal graph, we remove the causal influences from $y_1^{(k)}$ to $y_j^{(k)}$ for $j \geq 2$ by obtain projecting $x^{(k)}$ onto the orthocomplement to $\widehat{z}_1^{(k)}$, and the new causal graph can be viewed as a graph after marginalizing node 1, with only two nodes 2 and 3 and an edge $2 \to 3$ left. We then repeat these two steps to identify the topological ordering $1 \to 2$, $2 \to 3$, and $1 \to 3$. Since in the true graph, $1 \to 3$ is absent, a pruning step is thus needed. We denote the resulting estimates of $y_2^{(k)}$ and $y_3^{(k)}$ (resp. $z_2^{(k)}$ and $z_3^{(k)}$) as $\widetilde{y}_2^{(k)}$ and $\widetilde{y}_3^{(k)}$ (resp. $\widetilde{z}_2^{(k)}$ and $\widetilde{z}_3^{(k)}$). Here, for all $k \in [3]$, $\widetilde{y}_j^{(k)}$ depends on all $y_i^{(k)}$ for $i$ in the ancestors of $j$; whereas $\widehat{z}_j^{(k)}$ is $z_j^{(k)}$ up to scale and permutation.

In the pruning subroutine, we leverage a key observation that enables the detection of the spurious edge $1 \to 3$: Since in the true causal graph, $1 \to 3$ is absent, this will lead to certain rank-degeneracy of the coefficient matrices of regressing $\widehat{z}^{(k)}$ against $\widetilde{y}^{(k)}$, after concatenating over all environments. Repeating this across all edges, a pruned causal graph $\widehat{\mathcal{G}}$ can be obtained and $\widehat{\mathcal{G}} \sim_\pi \mathcal{G}$.

Since we do not have access to the ground truth or estimator up to scale and permutation for $y^{(k)}$, we regress $\widehat{z}^{(k)}$ against $\widetilde{y}^{(k)}$ and obtain unmixing matrices from the coefficient matrices to transform $\widetilde{y}^{(k)}$ to an estimator of $y^{(k)}$ up to $\sim_{\text{sur}}$

**Computing resources** All our experiments are conducted in one NVIDIA GeForce RTX 4090 GPU.

