# OpenReview forum: "Linear Causal Representation Learning by Topological Ordering, Pruning, and Disentanglement"
_ICLR.cc/2026/Conference — Submitted to ICLR 2026_

### Official Review · Reviewer_ba3u · 2025-10-29

**Soundness:** 3
**Presentation:** 3
**Contribution:** 2
**Rating:** 6
**Confidence:** 3

**Summary:**

This paper presents CREATOR, a novel algorithm for linear causal representation learning (CRL) from heterogeneous environments. The work focuses on a setting with a linear structural causal model over the latent variables and a linear mixing function mapping these latents to observations. The proposed CREATOR algorithm operates in three stages: inferring a topological ordering of the latent variables, pruning the resulting dense graph to identify the sparse causal DAG, and finally disentangling the latent features to the underlying causal DAG.

**Strengths:**

The main strength of this work lies in its successful relaxation of several restrictive assumptions made in the recent, closely related work of Jin & Syrgkanis (2024). Specifically, the proposed method does not require the distribution of exogenous noise to be identical across different environments, nor does it assume that different noise components within the same environment must have different distributions.

**Weaknesses:**

1. The manuscript's strong assumption of a linear model (linear SCM and mixing function) limits its applicability to complex, real-world scenarios which are often nonlinear.

2. The problem setup assumes all environments share the same observed variables, whereas in reality, environments may have only partially overlapping sets of observed variables. Considering this case would better reflect real-world conditions.

3. The manuscript is technically dense and lacks intuitive examples or explanations for its core mechanisms, making it difficult to follow.

4.  Similar to Jin & Syrgkanis (2024), a discussion comparing this work with the following papers is necessary:
    * A versatile causal discovery framework to allow causally-related hidden variables, ICLR, 2023.
    * Generalized independent noise condition for estimating causal structure with latent variables, JMLR, 2024.

5.  The manuscript claims that (line 345) it “yields a more efficient pruning procedure” than the method in Jin & Syrgkanis (2024). To substantiate this claim, an explicit efficiency comparison is needed, which would further enhance the contribution of the work.

**Questions:**

1. Lines 89–90 state that the algorithm can ``provably identify latent features and their causal mechanisms up to an equivalence class,'' while line 193 claims that ``the latent features and causal DAG can be uniquely recovered.'' This appears somewhat contradictory, as an equivalence class is not a unique DAG. Clarification would be appreciated.

---

> ### Author Response · Authors · 2025-11-21
> **Rebuttal part 1**
>
> # Rebuttal to the weaknesses part
>
> ## The manuscript's strong assumption of a linear model (linear SCM and mixing function) limits its applicability to complex, real-world scenarios which are often nonlinear.
>
> We appreciate the insightful suggestion. Extending our work to general structural causal models is a valuable direction for future research. While we acknowledge the limitations of linear models, we contend that they serve as a foundational first step before tackling more complex nonlinear problems, as supported by [1]. Moreover, we provide discussion about why such an extension can be non-trivial and have pointed out a possible direction of solving it in Appendix F of our manuscript.
>
> ## The problem setup assumes all environments share the same observed variables, whereas in reality, environments may have only partially overlapping sets of observed variables. Considering this case would better reflect real-world conditions.
>
> Thank you for raising this interesting open problem. It is a valuable research question to consider in future work. However, we want to first clarify that although we do not yet allow $x^{(k)}$ to have different ambient dimensions across environments, we do not assume that different directions of $x^{(k)}$ are aligned across environments. Further, we emphasize that in CRL or other invariant-based causal-inspired learning paradigms, it is important to impose certain invariance between different environments. To make theoretical progress, we assume that the mixing map $H$ is shared across environments. If we relax $x^{(k)}$ to have different dimensions but have partially overlapping directions, then each environment has its own mixing map $H^{(k)}$, but a submatrix $H$ of $H^{(k)}$ is shared across environments. Without imposing any additional assumptions on the remaining part of $H^{(k)}$ apart from $H$, it is impossible to obtain the same type of identifiability results as in our paper. When working out such an extension, the additional assumptions to be imposed need to be motivated from more concrete applications, which seems to deserve a separate paper.

---

> ### Author Response · Authors · 2025-11-21
> **Rebuttal part 2**
>
> ## The manuscript is technically dense and lacks intuitive examples or explanations for its core mechanisms, making it difficult to follow.
>
> Thank you for this suggestion for further clarifying the intuition of our proposed algorithm. Figure 1 in the submitted manuscript shows a 3-node example for Subroutine 1, which we were planning to use as an illustration of the intuition behind our algorithm.
>
> Nonetheless, we provide another toy model example to illustrate the main steps of our algorithm by adding the following paragraph in Appendix G. But if you have another suggestion, we are more than happy to incorporate it into our draft.
>
> "To better explain the intuition of our algorithm, in this section, we present an illustrative example. Consider a toy model in which $K = 3$ and $\forall k\in[3]$, $x^{(k)}= y^{(k)}$, $y^{(k)}=w^{(k)\top}y^{(k)}+z^{(k)}$. Here, we take both $H$ and $\Omega^{(k)}$ to be identities for simplicity, and $w^{(k)}$ is the weighted adjacency matrix of $\mathcal{G}$ representing the causal graph $1 \rightarrow 2$ and $2 \rightarrow 3$, where $1, 2, 3$ are vertex indices.
>
> In the first subroutine, we run ICA on $x^{(k)}$ for $k \in [3]$ and thus obtain the factorization $x^{(k)} = \hat{A}^{(k)} \hat{z}^{\dagger(k)}$, where $\hat{A}^{(k)}$ is a $3 \times 3$ matrix and $\hat{z}^{(k)}$ are independent components computed by ICA. As $H$ and $\mathcal{G}$ are invariant across environments, we can find a vector $\alpha_0$ such that $\alpha_0^\top x^{(k)} \propto_k y_r^{(k)}$ where $y_r^{(k)}$ corresponds to some root node, influenced only by the exogeneous noise $z_r^{(k)}$. Therefore, such $\alpha_0$ has to be parallel to one of the row vectors of all three unmixing matrices $\hat{A}^{(k)}$ for $k \in [3]$. A formal version of this claim can be found in Theorem 2.
>
> Next, we choose one row of all the rows in $\hat{A}^{(k)}$ across $k \in [3]$, denoted by $\hat{\alpha} _ 1$, such that for all $k \in [3]$ and $j \in [3]$, $\hat{z} _ 1 ^{(k)} := \hat{\alpha} _ 1 ^ \top x^{(k)}$ is independent of $r_ j ^{(k)}$, where $r_ j ^{(k)}$ is the residual of projecting $y_ j ^{(k)}$ onto $\hat{z}_ 1 ^{(k)}$. $\hat{\alpha} _ 1$ is then parallel to $\alpha_0$ because $r_ j ^{(k)}$ is independent of $\hat{z}_ 1 ^{(k)}$ if and only if $\hat{z}_ 1 ^{(k)}$ equals to a component of $z^{(k)}$ for all $k \in [3]$ up to scale by non-Gaussianity using the Darmois-Skitovitch theorem [4,5]. Furthermore, because the independence condition must be satisfied for all $k \in [3]$, $\hat{z}_ 1 ^{(k)}$ must also be equal to the root node in $y^{(k)}$ up to scale, by the non-degeneracy of $W^{(k)}$. Since $\hat{z}_ 1 ^{(k)}$ corresponds to the root node, $\tilde{y}_ 1 ^{(k)} := \hat{z}_ 1 ^{(k)}$ can serve as an estimator of $y_ 1 ^{(k)}$.
>
> Next, given that we have identified the root node in the original causal graph, we remove the causal influences from $y^{(k)} _ 1$ to $y^{(k)} _ j$ for $j\geq 2$ by obtain projecting $x^{(k)}$ onto the orthocomplement to $\hat{z}_ 1 ^{(k)}$, and the new causal graph can be viewed as a graph after marginalizing node $1$, with only two nodes $2$ and $3$ and an edge $2 \rightarrow 3$ left. We then repeat these two steps to identify the topological ordering $1 \rightarrow 2$, $2 \rightarrow 3$, and $1 \rightarrow 3$. Since in the true graph, $1 \rightarrow 3$ is absent, a pruning step is thus needed. We denote the resulting estimates of $y_ 2 ^{(k)}$ and $y_ 3 ^{(k)}$ (resp. $z_ 2 ^{(k)}$ and $z_ 3 ^{(k)}$) as $\tilde{y}_ 2 ^{(k)}$ and $\tilde{y}_ 3 ^{(k)}$ (resp. $\tilde{z}_ 2 ^{(k)}$ and $\tilde{z}_ 3 ^{(k)}$).  Here, for all $k \in [3]$, $\tilde{y}_ j ^{(k)}$ depends on all $y _ i^{(k)}$ for $i$ in the ancestors of $j$; whereas $\hat{z} _ j ^{(k)}$ is $z _ j ^{(k)}$ up to scale and permutation.
>
> In the pruning subroutine, we leverage a key observation that enables the detection of the spurious edge $1 \rightarrow 3$: Since in the true causal graph, $1 \rightarrow 3$ is absent, this will lead to certain rank-degeneracy of the coefficient matrices of regressing $\hat{z}^{(k)}$ against $\tilde{y}^{(k)}$, after concatenating over all environments. Repeating this across all edges, a pruned causal graph $\hat{\mathcal{G}}$ can be obtained and $\hat{\mathcal{G}}\sim_\pi \mathcal{G}$.
>
> Since we do not have access to the ground truth or estimator up to scale and permutation for $y^{(k)}$, we regress $\hat{z}^{(k)}$ against $\tilde{y}^{(k)}$ and obtain unmixing matrices from the coefficient matrices to transform $\tilde{y}^{(k)}$ to an estimator of $y^{(k)}$ up to $\sim _ {\mathrm{sur}}$.“

---

> ### Author Response · Authors · 2025-11-21
> **Rebuttal part 3**
>
> ## Similar to Jin \& Syrgkanis (2024), a discussion comparing this work with the following papers is necessary:
> - A versatile causal discovery framework to allow causally-related hidden variables, ICLR, 2023.
> - Generalized independent noise condition for estimating causal structure with latent variables, JMLR, 2024.
>
> Thanks for this suggestion. We will provide comparison between these two papers with our work in Introduction section by adding:
> ``Our work is similar to [1,2] in the sense that both investigate structure learning with latent variables. In [1,2], the causal graph is recovered up to Markov Equivalence Classes or perfectly recovered under certain additional assumptions on the graph structure, from observation data in one environment. However, our work leverages data from multiple environments and recovers the latent causal graph up to permutation without restrictions on the graph structure. Moreover, our algorithm can recover both the latent causal graph and latent features up to some equivalence class, while [1, 2] mainly focus on identifying the causal graph among observed and latent variables.''
>
> ## The manuscript claims that (line 345) it “yields a more efficient pruning procedure” than the method in Jin \& Syrgkanis (2024). To substantiate this claim, an explicit efficiency comparison is needed, which would further enhance the contribution of the work.
>
> Thank you for your advice. In both the pruning subroutine in our manuscript and the Identify-Parents algorithm in Jin \& Syrgkanis (2024), SVD is conducted. In Jin \& Syrgkanis (2024), dimension of space spanned by $K$ vectors in $\mathbb{R}^d$ is computed through SVD for each possible edge while in our pruning subroutine, we compute the rank of $\hat{C} _ {i,j}\in\mathbb{R}^{K\times (i-j)}$ and $\tilde{C} _ {i,j}\in\mathbb{R}^{K\times (i-j+1)}$ for each possible edge from node $i$ to node $j$ with $i\geq j+1$.
> Concretely, in the pruning step, we regress $\hat{z}^{(k)}$ against $\tilde{y}^{(k)}$ and denote the regression coefficient as $\hat{B}^{(k)} \in \mathbb{R}^{d \times d}$. For any different $1 \leq j \leq i - 1 \leq d$, we construct $\mathbb{R}^{K} \ni \hat{B}_ {i,j} \coloneqq (\hat{B}^{(k)}_ {i,j} , k\in[K])$ and $\mathbb{R}^{K \times (i - j)} \ni \hat{C} _ {i,j}\coloneqq (\hat{B} _ {i,l},l\in\{j+1,\dots,i\})$ and $\tilde{C} _ {i,j} \coloneqq (\hat{B} _ {i,j},\hat{C} _ {i,j})$. Next, we compute the rank of $\hat{C} _ {i,j}$ and $\tilde{C} _ {i,j}$ and if and only if $\mathrm{rank} (\hat{C} _ {i,j}) = \mathrm{rank}(\tilde{C} _ {i,j}) - 1$, we conclude that $j \in \mathrm{pa} (i)$.
> As $\tilde{C}_{i,j}=\hat{C} _ {i,j+1}$, we actually need to compute only $\tilde{C} _ {i,j}$ for $j\geq 2$. In this regard, we claim that our pruning method is more efficient. Since this is not really the main advantage of our algorithm is not the efficiency in this aspect, we plan to only discuss this point in Appendix C.
>
> # Rebuttal to the questions part
>
> ## Lines 89–90 state that the algorithm can provably identify latent features and their causal mechanisms up to an equivalence class,'' while line 193 claims that the latent features and causal DAG can be uniquely recovered.'' This appears somewhat contradictory, as an equivalence class is not a unique DAG. Clarification would be appreciated.
>
> Thank you for pointing this out! We now change line 191-193 to "Consistent with much of the current CRL literature, we focus on identifiability -- namely, whether the latent features and causal DAG can be recovered up to $\sim _ {\mathrm{sur}}$ in the limit of infinite data, and whether our proposed procedure achieves this recovery in theory."
>
> __Reference__
>
> [1] A Versatile Causal Discovery Framework to Allow Causally-Related Hidden Variables. Xinshuai Dong, Biwei Huang, Ignavier Ng, Xiangchen Song, Yujia Zheng, Songyao Jin, Roberto Legaspi, Peter Spirtes, Kun Zhang.
>
> [2] Generalized Independent Noise Condition for Estimating Causal Structure with Latent Variables. Feng Xie, Biwei Huang, Zhengming Chen, Ruichu Cai, Clark Glymour, Zhi Geng, Kun Zhang.
>
> [3] Learning causal representations from general environments: Identifiability and intrinsic ambiguity. Jikai Jin and Vasilis Syrgkanis.
>
>
> [4] Analyse g{\'e}n{\'e}rale des liaisons stochastiques: etude particuli{\`e}re de l'analyse factorielle lin{\'e}aire. Darmois, George.
>
> [5] On a property of the normal distribution. Skitovitch, Viktor P.

---

> > ### Comment · Area_Chair_LBcr · 2025-11-28
> >
> > Dear Reviewer,
> >
> > Please make sure you read the authors' response and engage with them in the discussion before the end of the discussion period on **Dec 03 '25 09:00 PM UTC**. This is a hard deadline.
> >
> > Thank you for supporting quality peer review at ICLR.
> >
> > AC

---

### Official Review · Reviewer_xFXr · 2025-11-02

**Soundness:** 3
**Presentation:** 2
**Contribution:** 2
**Rating:** 4
**Confidence:** 3

**Summary:**

The paper introduces CREATOR, a new algorithm for linear causal representation learning (CRL) designed to relax some of the assumptions made in LiNGCReL (Jin & Syrgkanis, 2024).
The model assumes a linear structural causal model over latent variables and a linear mixing to observations. The method consists of three stages: (i) ordering and feature recovery using independence criteria, (ii) DAG pruning based on rank analysis across multiple environments, and (iii) feature disentanglement to recover latent variables up to a structural equivalence class.
Theoretical claims are supported by identifiability proofs relying on non-Gaussianity and independence arguments, and empirical validation is provided on small synthetic datasets and a qualitative case study on large language models (LLMs).

**Strengths:**

* The paper is well structured and formally presented. The proposed model formulation is clear and consistent with recent linear CRL frameworks such as those by Squires et al. (2023) and Jin & Syrgkanis (2024).
* The attempt to connect linear CRL with representation analysis in large language models is conceptually appealing and relevant to ongoing discussions on the linear representation hypothesis (Arora et al., 2016; Park et al., 2024).
* The relaxation of environment-specific assumptions, allowing heterogeneous noise distributions, addresses a practical limitation of earlier models and enhances the potential applicability of the method.

**Weaknesses:**

* **Assumption relaxation and identifiability.**
  The claim of weaker assumptions is debatable. Although CREATOR removes the requirement of identical noise across environments, it still depends on strong non-Gaussianity and independence conditions, similar to those used in LiNGAM-based approaches. The improvement over LiNGCReL thus appears more incremental rather than fundamental.

* **Dependence on ICA and independence testing.**
  The algorithm’s first stage heavily relies on ICA and associated independence measures, meaning the identifiability largely stems from ICA theory rather than a novel mechanism. In addition, the theoretical condition for independence is replaced in practice by the HSIC statistic (Gretton et al., 2005), which lacks formal consistency guarantees and is known to be unstable in empirical causal discovery (Rolland et al., 2022).

* **Fragility of pruning and disentanglement stages.**
  To the best of my understanding, the rank-based pruning assumes exact linear independence across environments and may be highly sensitive to small perturbations or finite-sample noise. This can propagate errors into the disentanglement stage, resulting in cascading inaccuracies, a problem noted in previous multi-stage causal methods (Varıcı et al., 2024b). Competing approaches such as Buchholz et al. (2023) adopt more robust regularization strategies.

* **Empirical limitations.**
  The synthetic experiments are limited to low-dimensional data (d ≤ 7, n = 1000) and only compared to LiNGCReL, without evaluation against broader baselines such as Squires et al. (2023) or nonlinear CRL models. The LLM case study is based on a pre-defined DAG and qualitative interpretation of latent “concepts,” making causal conclusions difficult to validate. Moreover, no runtime, ablation, or stability analyses are reported.

* **Presentation and clarity.**
  Some notation, such as the equivalence relations ($\sim \pi$, $\sim \Delta$, $ \sim sur$) is introduced formally but lacks intuitive explanation. Including a small illustrative example (e.g., a 3-node toy system as in Ahuja et al., 2023) would improve readability.

**Questions:**

* Could the authors clarify in what specific sense Assumptions 1–3 are weaker than those in Jin & Syrgkanis (2024)? Does the notion of identifiability here correspond to ancestral, Markov, or exact equivalence?
* How is the HSIC test implemented in practice (kernel choice, thresholds), and how sensitive is the ordering step to deviations from true independence?
* Could disentanglement quality be assessed with additional metrics, such as mutual information or subspace similarity, beyond LocR²?
* In the LLM study, how do the authors verify that the assumed DAG reflects causal rather than syntactic relations? Would it be possible to compare the recovered features to known linguistic embeddings for validation?

---

> ### Author Response · Authors · 2025-11-21
> **Rebuttal part 1**
>
> # Rebuttal to the weaknesses part
> ## __Assumption relaxation and identifiability.__ The claim of weaker assumptions is debatable. Although CREATOR removes the requirement of identical noise across environments, it still depends on strong non-Gaussianity and independence conditions, similar to those used in LiNGAM-based approaches. The improvement over LiNGCReL thus appears more incremental rather than fundamental.
>
> We appreciate this criticism. In fact, we definitely agree with reviewer xFXr that even weaker assumptions than what we assume are desirable. But for CRL, the structure of the problem is not yet clear to us even under the ``strong'' assumptions of non-Gaussianity and independent exogenous noises, let alone the even more difficult setting recommended in the comment.
>
> Specifically, the distribution assumptions that we relaxed over the noise variables are critical to the success of LiNGCReL. Relaxing the original assumptions for LiNGCReL to work makes linear CRL closer to real-world conditions. For instance, data collected from different labs or experimentation equipments could have different types of noises, due to the heterogeneity in measurement devices. Yet noise components within one environment could be more likely to share the same distribution, because the measurements are presumably recorded using the same device or under a common environmental condition. Thus allowing the possibility of the same noise distribution within one environment can be useful.
>
> On the other hand, independence among components of noise variables and noise variables across environments is commonly assumed in CRL studies [1,2,3] and very important for discovering the underlying causal structure of the latent variables. Consider a toy model in which $x^{(k)}=y^{(k)}$ and $y^{(k)}=w^{(k)}y^{(k)}+z^{(k)}$ where $z^{(k)}_1$ and $z^{(k)}_2$ are correlated. Here $y^{(k)}_1$ and $y^{(k)}_2$ are correlated. However, without assuming independence between $z^{(k)}_1$ and $z^{(k)}_2$, we cannot further distinguish the causal structure of $y^{(k)}$.
>
> For the multi-environment linear CRL problem we considered, it is rather difficult to identify the latent causal variables up to $\sim_{\mathrm{sur}}$ for Gaussian noise variables. Again consider a toy model with two latent variables such that $x^{(k)}=y^{(k)}$ and $y^{(k)}_1$ and $y_2^{(k)}$ are independent and their variance are both 1. In this case, $\hat{y}^{(k)}=(y^{(k)}_1-y^{(k)}_2,y^{(k)}_1-y^{(k)}_2)^\top$ and $x^{(k)}=((\hat{y}^{(k)}_1+\hat{y}^{(k)}_2)/2,(\hat{y}_2^{(k)}-\hat{y}^{(k)}_1)/2)^\top$ is a solution to the CRL problem but $\hat{y}^{(k)}$ is not identified up to $\sim _ {\mathrm{sur}}$ due to the Gaussianity.
>
> ## __Dependence on ICA and independence testing.__ The algorithm’s first stage heavily relies on ICA and associated independence measures, meaning the identifiability largely stems from ICA theory rather than a novel mechanism. In addition, the theoretical condition for independence is replaced in practice by the HSIC statistic (Gretton et al., 2005), which lacks formal consistency guarantees and is known to be unstable in empirical causal discovery (Rolland et al., 2022).
>
> Thank you for raising this concern. In our opinion, the actual key step of the first subroutine of our algorithm is not ICA, but to discover a root-node component of $y^{(k)}$ (latent features) by solving the following optimization problem: $\hat{\alpha} _ {i} \coloneqq \mathop{\arg\min} \limits_{\alpha} L^{(i)} (\alpha,x) \coloneqq \sum\limits_{k=1}^K \sum_{j=1}^d \mathrm{MI} (\alpha^\top\mathrm{proj}_i{x}^{(k)},r^{(k)}_j)$ according to Theorem 2. Since any solution $\hat{\alpha}$ to this optimization problem satisfies that $\hat{\alpha}^\top \mathrm{proj}_i{x}^{(k)}$ is a component of $z^{(k)}$, adding a constraint such that $\alpha$ is one of the row vectors of the unmixing matrices estimated by any ICA algorithm can simplify the optimization problem. Thus at least in our setup, ICA is only a shortcut to realize the underlying mathematical program, not necessarily a fundamental building block.
>
> In Theorem 2 of [4], a uniform convergence bound on HSIC estimator using positive definite kernel is provided and we also derive a convergence analysis for our algorithm in Appendix D.1 of our manuscript.
>
> In terms of the instability of the HSIC statistic in empirical causal discovery problem, we did not find a relevant part in Rolland et al., 2022, as suggested by Reviewer xFXr. We are more than willing to further discuss this issue if the reviewer could point to us the specific part of Rolland et al.'s paper or any relevant references. We did not find that the procedure suffers from numerical instability empirically, at least in our settings.

---

> ### Author Response · Authors · 2025-11-21
> **Rebuttal part 2**
>
> ## __Fragility of pruning and disentanglement stages.__ To the best of my understanding, the rank-based pruning assumes exact linear independence across environments and may be highly sensitive to small perturbations or finite-sample noise. This can propagate errors into the disentanglement stage, resulting in cascading inaccuracies, a problem noted in previous multi-stage causal methods (Varıcı et al., 2024b). Competing approaches such as Buchholz et al. (2023) adopt more robust regularization strategies.
>
> Thank you for your advice. As mentioned in (Varıcı et al., 2024b), errors from early subroutines can propagate to later ones. Given this potential for error propagation, a further analysis of the influence of the pruning step on the disentanglement step is necessary. In our implementation, we conduct SVD on $\hat{C} _ {i,j}$ and $\tilde{C}_{i,j}$ and report the number of singular value greater than some threshold $\tau$ as the rank. For $d=3$, we chose $\tau\coloneqq 0.01$. For $d>3$, we chose $\tau\coloneqq \frac{5}{\sqrt{d}}$ when $K=d$ and $\tau\coloneqq \frac{3\sqrt{K}}{d}$ when $K=2d$. From the experiment results, the propagate errors does not cause unacceptable inaccuracies in our situation. To better illustrate the instability in the pruning step induced by the threshold, we provide the results for different choices of $\tau$ in the following tables:
>
> ### Algorithm performance for $K=d$ and $d=5$ with different threshold $\tau$
>
> | $\tau$ | 0.1  | 0.4  | 0.7  | 1.0  | 1.3  | 1.6  | 1.9  | 2.2  | 2.5  | 2.8  |
> |--------|------|------|------|------|------|------|------|------|------|------|
> | SHD    | 3.1  | 2.5  | 2.5  | 2.7  | 2.8  | 3    | 3.3  | 3.5  | 2.8  | 3    |
> | $\mathrm{LocR}^2$ | 0.69 | 0.68 | 0.72 | 0.74 | 0.70 | 0.67 | 0.71 | 0.73 | 0.71 | 0.71 |
>
>
> ### Algorithm performance for $K=2d$ and $d=5$ with different threshold $\tau$
>
> | $\tau$ | 0.1  | 0.4  | 0.7  | 1.0  | 1.3  | 1.6  | 1.9  | 2.2  | 2.5  | 2.8  |
> |--------|------|------|------|------|------|------|------|------|------|------|
> | SHD    | 3.5  | 2.5  | 2.2  | 2.0  | 2.0  | 2.0  | 2.1  | 2.7  | 2.8  | 3.2  |
> | $\mathrm{LocR}^2$ | 0.75 | 0.77 | 0.84 | 0.87 | 0.81 | 0.81 | 0.82 | 0.77 | 0.78 | 0.74 |
>
> We are open to the reviewer's suggestions for improving the pruning and disentanglement subroutines to reduce errors. Moreover, the regularizer proposed by Buchholz et al. (2023) is not directly applicable to our method, since the adjacency matrix of the causal graph in their work is parameterized and learned through the optimization of a loss function—representing a different methodological framework for causal structure learning.

---

> > ### Author Response · Authors · 2025-11-21
> > **Rebuttal part 3**
> >
> > ## __Empirical limitations.__ The synthetic experiments are limited to low-dimensional data ($d \leq 7$, $n = 1000$) and only compared to LiNGCReL, without evaluation against broader baselines such as Squires et al. (2023) or nonlinear CRL models. The LLM case study is based on a pre-defined DAG and qualitative interpretation of latent “concepts,” making causal conclusions difficult to validate. Moreover, no runtime, ablation, or stability analyses are reported.
> >
> > We agree that evaluating our method in higher-dimensional settings is more desirable. Our algorithm is designed to be scalable to higher dimensions. The choice of evaluating our method with latent dimensions $d \in$ \{3,5,7,9\} for the case of $K = d$, and $d \in$ \{2,3,5,7\} for $K = 2d$, was primarily motivated due to the evaluation part, which is not relevant to the algorithm when deploying it in practice. The evaluation requires computing $\frac{1}{dK}\sum\limits_{i=1}^d\sum\limits_{k=1}^K\mathrm{LocR}_{i,k}^2$ across all permutations. When $d$ gets higher, this evaluation step is the most time-consuming one. For example, when $d = 11$, there are $11!$ permutations in total to conduct the evaluation. Therefore, in our submitted draft, we adhered to the experimental settings established by Jin \& Syrgkanis, concentrating on scenarios with lower latent dimensions in both our synthetic and real-data analyses. The relatively small latent dimension also occurs in several recent papers on CRL with real-world applications [5, 6] (though their settings are different from ours). Besides, scenarios with high-dimensional observation data are very common and thus we also evaluated our methods with slightly higher ambient dimension $p = 5 d, 100, 200$ under Setting 1 and the results can be shown as follows (though with the same choice of $d$'s). The results are largely similar to smaller $p = d$ in the submitted draft (and not surprisingly, larger $p$ leads to minor degradation in the performance):
> >
> > ### evaluation for ambient dimension higher $p$ with $K=d$ in setting 1
> > |          | SHD ($p=5d$) | $\mathrm{LocR}^2$ ($p=5d$) | SHD ($p=100$) | $\mathrm{LocR}^2$ ($p=100$) | SHD ($p=200$) | $\mathrm{LocR}^2$ ($p=200$) |
> > |----------|-------------|----------------------------|--------------|----------------------------|--------------|----------------------------|
> > | $d=3$    | 0.2         | 0.88                       | 0.0          | 0.88                       | 0.1          | 0.81                       |
> > | $d=5$    | 2.9         | 0.69                       | 2.5          | 0.55                       | 2.6          | 0.59                       |
> > | $d=7$    | 4.3         | 0.63                       | 4.7          | 0.52                       | 4.2          | 0.45                       |
> > | $d=9$    | 6.6         | 0.52                       | 7.3          | 0.39                       | 7.0          | 0.39                       |
> >
> > ### evaluation for ambient dimension higher $p$ with $K=2d$ in setting 1
> > |          | SHD ($p=5d$) | $\mathrm{LocR}^2$ ($p=5d$) | SHD ($p=100$) | $\mathrm{LocR}^2$ ($p=100$) | SHD ($p=200$) | $\mathrm{LocR}^2$ ($p=200$) |
> > |----------|---------------|-----------------------------|---------------|-----------------------------|---------------|-----------------------------|
> > | $d=2$    | 0.0           | 0.99                        | 0.0           | 0.85                        | 0.0           | 0.94                        |
> > | $d=3$    | 0.0           | 0.99                        | 0.0           | 0.78                        | 0.0           | 0.87                        |
> > | $d=5$    | 1.9           | 0.75                        | 3.0           | 0.71                        | 2.7           | 0.66                        |
> > | $d=7$    | 4.7           | 0.59                        | 6.5           | 0.53                        | 4.9           | 0.46                        |
> >
> > In Squires et al. [7], perfect single-node intervention is assumed, which is different from our general environments setting. Therefore, it is not directly comparable to our method. To the best of our knowledge, there is no non-linear CRL algorithm developed for our setting and thus we did not compare against nonlinear models. In our LLM case study, we focus on finding the latent causal relationship among background, condition and ending. We build the ground truth for causal graph based on common knowledge and this causal graph is also mentioned in [7].
> >
> > We provide the runtime for $K=d$ and $K=2d$ in the following tables:
> >
> > ### Runtime for $K=d$
> > | d   | 3    | 5    | 7     | 9     |
> > |-----|------|------|-------|-------|
> > | t   | 0.68 | 4.60 | 17.20 | 45.21 |
> >
> > ### Runtime for $K=d$
> > | d   | 2    | 3    | 5     | 7     |
> > |-----|------|------|-------|-------|
> > | t   | 0.43 | 2.37 | 18.02 | 67.36 |
> >
> > (Please see the following comment for the rest.)

---

> ### Author Response · Authors · 2025-11-21
> **Rebuttal part 4**
>
> (This comment continues from Rebuttal part 3)
>
> As the only hyperparameter used in this work is the threshold for rank estimation in the pruning subroutine, we provide the sensitivity analysis in the following tables:
>
> ### Algorithm performance for $K=d$ and $d=5$ with different threshold $\tau$
>
> | $\tau$ | 0.1  | 0.4  | 0.7  | 1.0  | 1.3  | 1.6  | 1.9  | 2.2  | 2.5  | 2.8  |
> |--------|------|------|------|------|------|------|------|------|------|------|
> | SHD    | 3.1  | 2.5  | 2.5  | 2.7  | 2.8  | 3    | 3.3  | 3.5  | 2.8  | 3    |
> | $\mathrm{LocR}^2$ | 0.69 | 0.68 | 0.72 | 0.74 | 0.70 | 0.67 | 0.71 | 0.73 | 0.71 | 0.71 |
>
>
> ### Algorithm performance for $K=2d$ and $d=5$ with different threshold $\tau$
>
> | $\tau$ | 0.1  | 0.4  | 0.7  | 1.0  | 1.3  | 1.6  | 1.9  | 2.2  | 2.5  | 2.8  |
> |--------|------|------|------|------|------|------|------|------|------|------|
> | SHD    | 3.5  | 2.5  | 2.2  | 2.0  | 2.0  | 2.0  | 2.1  | 2.7  | 2.8  | 3.2  |
> | $\mathrm{LocR}^2$ | 0.75 | 0.77 | 0.84 | 0.87 | 0.81 | 0.81 | 0.82 | 0.77 | 0.78 | 0.74 |
>
> We have actually conducted an ablation study in Appendix E.3, showing the influence that the topological inference step has on the performance of our algorithm.
>
> ## __Presentation and clarity.__ Some notation, such as the equivalence relations ($\sim _ \pi$, $\sim _ {\triangle}$, $\sim_{\mathrm{sur}}$) is introduced formally but lacks intuitive explanation. Including a small illustrative example (e.g., a 3-node toy system as in Ahuja et al., 2023) would improve readability.}
>
> Thank you for your advice. We will provide a illustrative toy example in Appendix G by adding that
>
> “Consider a toy model such that for $\forall k\in[3]$, $x^{(k)}=Hy^{(k)}$, $y^{(k)}=w^{(k)\top}y^{(k)}+\Omega^{(k)}z^{(k)}$ where $H=I$, $w^{(k)}$ is weighted adjacency matrix of $\mathcal{G}$ with two edges including $1 \rightarrow 2$ and $2 \rightarrow 3$, where $1, 2, 3$ denotes the node indices. When $\hat{y}^{(k)}\sim_\pi y^{(k)}$ for $k\in[3]$, there exist a permutation $\pi$ and three non-zero number $\lambda_1$, $\lambda_2$ and $\lambda_3$ such that $\hat{y}_i^{(k)}=\lambda_iy _ {\pi(i)}^{(k)}$ for any $i\in[3]$ for all $k\in[3]$.
>
> When $\hat{y}^{(k)}\sim_ \triangle y^{(k)}$ for $k\in[3]$, there exist a lower triangular matrix $B$ and a permutation $\pi$ such that $\hat{y} _ 1^{(k)}=B _ {1,1}y_ {\pi(1)}^{(k)}$,\ $\hat{y} _ 2^{(k)}=B _ {2,1}y_{\pi(1)}^{(k)}+B_{2,2}y_{\pi(2)}^{(k)}$, and
>     $\hat{y} _ 3^{(k)}=B_{3,1}y_{\pi(1)}^{(k)}+B_{3,2}y_{\pi(2)}^{(k)}+B_{3,3}y_{\pi(3)}^{(k)}$ for all $k\in[3]$. According to the definition of surrounding set, we have $\overline{\mathrm{sur}} _ \mathcal{G}(1)=\{1\}$, $\overline{\mathrm{sur}} _ \mathcal{G}(2)=\{2\}$ and $\overline{\mathrm{sur}} _ \mathcal{G}(3)=\{2,3\}$. When $\hat{y}^{(k)}\sim _ \mathrm{sur} y^{(k)}$ for $k\in[3]$, there exist a lower triangular matrix $B$ and a permutation $\pi$ such that
>     $\hat{y} _ i^{(k)}=B_{i,i}y_{\pi(i)}^{(k)}$ ($i\in$\{1,2\}), and $\hat{y}^{(k)} _ 3=B_{3,2}y_{\pi(2)}^{(k)}+B_{3,3}y_{\pi(3)}^{(k)}$   .“

---

> > ### Author Response · Authors · 2025-11-21
> > **Rebuttal part 5**
> >
> > # Rebuttal to the questions part
> >
> > ## Could the authors clarify in what specific sense Assumptions 1–3 are weaker than those in Jin \& Syrgkanis (2024)? Does the notion of identifiability here correspond to ancestral, Markov, or exact equivalence?
> >
> > Assumptions 1–3 are adopted directly from [1]. The key assumption relaxed in our work is that the exogenous noise variables are not required to be identically distributed across diverse environments, nor are their components constrained to follow different distributions within the same environment. In our manuscript, we provide identifiability of latent feature $y^{(k)}$ and causal graph $\mathcal{G}$ up to $\sim _ \mathrm{sur}$, which is defined:
> >
> > there exists a permutation $\pi$ on $[d]$ and a lower triangular matrix $B$ where for $\forall j\in[d],i\notin\overline{\mathrm{sur}}(j)$, $B_{i, j}=0$, such that the following holds:
> > - $\forall \, i,j\in[d]$, $i\in \mathrm{pa}(j)\iff \pi(i)\in \mathrm{pa}(\pi(j))$;
> > - $\hat{y}^{(k)} = B P_\pi y^{(k)}$, where $P_\pi$ denotes the permutation matrix corresponding to $\pi$.
> >
> >
> > ## How is the HSIC test implemented in practice (kernel choice, thresholds), and how sensitive is the ordering step to deviations from true independence?
> >
> > Thank you for your question. The HSIC test is used in the first subroutine and to select an $\alpha_i$ such that $\alpha_i^\top x^{(k)}$ is independent of $x^{(k)}-\mathbb{E}(x^{(k)}|\alpha_i^\top x^{(k)})$ for any $k\in[K]$. In our implementation, we choose Gaussian kernel such that $k(x,y)=\mathrm{exp}(\frac{-||x-y||^2}{2\sigma^2})$. After ICA, we simply search over all $K \cdot d$ row vectors from all $K$ unmixing matrices and identify $\hat{\alpha}_i$ as the the row vector that minimizing the HSCI estimator. Therefore, we do not use any thresholds in this procedure.
> >
> > For sensitivity of the ordering step to deviations from true independence, we present the evaluation of inferred topological ordering with topological order divergence [4] defined as $D(\pi,A)=\sum\limits_{i=1}^d\sum\limits_{j:\pi_i>\pi_j}A_{i,j}$ where $A$ denotes the adjacency matrix of $\mathcal{G}$. From the results, we can see that ordering step is quite accurate for the independence criterion in our implementation.
> >
> > ### Topological ordering performance for $K=d$
> >
> > | $d$       | 3   | 5   | 7   | 9   |
> > |-----------|-----|-----|-----|-----|
> > | $D(\pi,A)$ | 0.2 | 0.3 | 0.8 | 1.1 |
> >
> > ### Topological ordering performance for $K=2d$
> >
> > | $d$       | 2   | 3   | 5   | 7   |
> > |-----------|-----|-----|-----|-----|
> > | $D(\pi,A)$ | 0.0 | 0.1 | 0.4 | 0.3 |
> >
> > ## Could disentanglement quality be assessed with additional metrics, such as mutual information or subspace similarity, beyond $\mathrm{LocR}^2$?
> > Since the latent features are identified up to $\sim_\mathrm{sur}$, for any node $i$ with non-empty surrounding set, $\hat{y}^{(k)}_i$ is still entangled with \{$y^{(k)} _ j:j\in\overline{\mathrm{sur}}(i)$\}. Thus mutual information is not an appropriate metric for our work. Furthermore, subspace similarity is also inadequate. This is because we address linear causal representation learning (CRL) with a linear mapping from latent features to observed data. When $p = d$, even the raw observational data $x^{(k)}$ can minimize the subspace similarity measure $\sin\theta(Y,V) = |(I - YY^\top)V|_2$ introduced in [5]. Given that the estimated latent features in both our method and that of Jin \& Syrgkanis (2024) are derived via a linear transformation of the observational data, the resulting values of this metric are consistently below $10^{-4}$ in both cases. Thus, this metric is ill-suited for evaluating the performance of our algorithm.
> >
> > Ideally, we can only identify $y^{(k)} _ i$ as $\hat{y}^{(k)} _ i=\sum\limits _ {j\in\overline{\mathrm{sur}}(i)}a_{ij}y_j$, and $\mathrm{LocR}^2$ shows on average how well $\hat{y}^{(k)}_i$ can be regressed on $\{y^{(k)}_j,j\in\overline{sur}(i)\}$ for all $i\in[d]$. Consequently, $\mathrm{LocR}^2$ alone suffices for evaluating our algorithm.

---

> ### Author Response · Authors · 2025-11-21
> **Rebuttal part 6**
>
> ## In the LLM study, how do the authors verify that the assumed DAG reflects causal rather than syntactic relations? Would it be possible to compare the recovered features to known linguistic embeddings for validation?
>
> Thank you for this excellent and insightful suggestion. The idea of validating our recovered causal features against established linguistic embeddings is intuitively appealing and reflects a deep interest in model interpretability.
>
> However, accurately extracting the exact textual content corresponding to background, condition, or outcome from the full text remains challenging. As an alternative, only a limited set of keywords can be obtained as proxies for these variables. In contrast, the latent features are derived from observed data—specifically, from the final hidden states of selected large language models (LLMs) when the entire text is provided as input. We therefore argue that validation based solely on such proxy variables is inadequate. Instead, we identify a permutation over the latent features that minimizes the prediction error of these features when using the proxy variables, by training a neural network. This approach yields latent features with a meaningful and interpretable ordering.
>
> __Reference__
>
> [1] Learning causal representations from general environments: Identifiability and intrinsic ambiguity. Jikai Jin and Vasilis Syrgkanis.
>
> [2] Linear causal representation learning from unknown multi-node interventions. Burak Varıcı, Emre Acartürk, Karthikeyan Shanmugam, and Ali Tajer.
>
> [3] General Identifiability and Achievability for Causal Representation Learning. Burak Varıcı and Emre Acartürk and Karthikeyan Shanmugam and Ali Tajer.
>
> [4] Recovery guarantee of weighted low-rank approximation via alternating minimization. Yuanzhi Li, Yingyu Liang, Andrej Risteski.
>
> [5] CausalVAE: Disentangled Representation Learning via Neural Structural Causal Models. Mengyue Yang and Furui Liu and Zhitang Chen and Xinwei Shen and Jianye Hao and Jun Wang.
>
> [6] Learning Causally Disentangled Representations via the Principle of Independent Causal Mechanisms. Aneesh Komanduri and Yongkai Wu and Feng Chen and Xintao Wu.
>
> [7] Beyond What If: Advancing Counterfactual Text Generation with Structural Causal Modeling. Ziao Wang, Xiaofeng Zhang, Hongwei Du.

---

> > ### Comment · Reviewer_xFXr · 2025-11-26
> >
> > I thank the authors for their response and for the additional experiments and clarifications provided. Most of my concerns were addressed, therefore I raised my score.
> >
> > **HSIC**: For the HSIC stability point raised, I apologize as I included the wrong reference: I was referring to [a]. But my concerns on this are now solved as I misunderstood the HSIC is actually used just as score for selecting $\\hat{\alpha}_i$ and not directly minimized.
> >
> > _[a] Mooij, Joris M., et al. "Distinguishing cause from effect using observational data: methods and benchmarks." Journal of Machine Learning Research_
> >
> >
> > **Scalability**: One point I'm still not totally convinced about the scalability of the experiments on the number of latent dimensions (causal variables): if the bottleneck is in the evaluation, rather than evaluating if the graph is recovered correctly, could it be possible to evaluate on some downstream task that is assumed to depend on true graph? Would this allow to scale to higher dimensions and more realistic settings?  I understand this is these low dimensional setting is the common status of the literature and I'm not asking to perform additional experiments, but I think a discussion on this would be valuable for the general machine learning audience to understand how applicable is the proposed method.

---

> > > ### Author Response · Authors · 2025-11-27
> > >
> > > Thank you for kindly raising our score and the suggestion for new direction of evaluation. In fact, we agree that it is an important research problem for the CRL literature. We will add the following discussion in the last section of the main text:
> > >
> > > ”We believe that evaluating the performance of CRL algorithms is an underexplored research area in the CRL literature. In real life applications, due to the unavailability of ground truth and the computational difficulty of evaluation even if ground truth is given, a reasonable approach is to evaluate the performance of downstream tasks of CRL [1]. For example, an important application of CRL is to leverage the underlying causal graph structure and intervene latent features to achieve certain desired changes in the observed data (such as natural images, sounds, and etc.) [2]. Whether the desired changes manifest after intervening the latent features can also serve as an indirect evidence of the success of CRL algorithms.“
> > >
> > > __Reference__
> > >
> > > [1] Structural intervention distance for evaluating causal graphs. Peters, Jonas, and Peter Bühlmann.
> > >
> > > [2] CausalVAE: Disentangled Representation Learning via Neural Structural Causal Models. Mengyue Yang and Furui Liu and Zhitang Chen and Xinwei Shen and Jianye Hao and Jun Wang.

---

### Official Review · Reviewer_rpXZ · 2025-11-02

**Soundness:** 3
**Presentation:** 3
**Contribution:** 2
**Rating:** 6
**Confidence:** 4

**Summary:**

The paper introduces CREATOR (Causal REpresentation leArning via Topological Ordering, Pruning, and Disentanglement), a linear causal representation learning (CRL) method that assumes (i) a linear SCM over latent variables and (ii) a linear mixing from latents to observations. Relative to prior linear-CRL work, the method weakens distributional assumptions by requiring only non-Gaussian, independent noise (≤1 Gaussian component) and allows noise distributions to vary across environments. Synthetic experiments show improved LocR² (latent recovery) and SHD (DAG accuracy) versus the previous LiNGCReL algorithm; a small LLM case study illustrates feasibility under the “linear representation hypothesis.”

**Strengths:**

- Assumption 1 permits environment-specific noise distributions and only requires non-Gaussian independent components ($\leq 1$ Gaussian), and Theorem 1 establishes identifiability up to ∼sur with ≥ d environments. This is a meaningful step beyond prior linear-CRL assumptions.

- On synthetic tasks across $d\in\{2,3,5,7\}$ and $K\in\{d,2d\}$, CREATOR improves LocR² and SHD over LiNGCReL (Fig. 2; Fig. 3 in Appendix).

- LLM case study is neat as a proof-of-concept.

**Weaknesses:**

- While improving the performance of LiNGCReL, it appears that the underlying idea of the identification algorithm is largely the same, limiting the novelty of this paper.

**Questions:**

- How does CREATOR behave when d is under- or over-estimated?

- In noisy finite samples, do you use singular-value thresholds or bootstrap ranks for deciding the 1-rank drop?

---

> ### Author Response · Authors · 2025-11-21
>
> # Rebuttal to the weaknesses part
> ##  While improving the performance of LiNGCReL, it appears that the underlying idea of the identification algorithm is largely the same, limiting the novelty of this paper.
> Thank you for your comment. First, we would like to emphasize that our identification algorithm is not largely the same as LiNGCReL, although the first step of both involves ICA. Compared to LiNGCReL, our algorithm is developed under relaxed assumptions: (1) the distributions of noise variables $z^{(k)}$ across environments are allowed to be different and (2) the distributions of different components of exogenous noise variables $z^{(k))}$ (of dimension $d$) within one environment are no longer restricted to be different from each other. To this end, we introduce a new subroutine for topological ordering inference. This innovation yields two critical advances: (1) it facilitates the alignment of exogenous noises recovered from multiple environments under the aforementioned more relaxed assumptions than those for LiNGCReL, and (2) it supplies the topological ordering information that is utilized by the two subsequent subroutines. The ablation study also shows that accurately recovering the topological information is a key reason for the good performance of our algorithm. However, the procedure in LiNGCReL fails to identify the mixing function, latent features, and the DAG up to the same equivalence classes under our relaxed assumptions.
>
> # Rebuttal to the questions part
> ## How does CREATOR behave when d is under- or over-estimated?
> When $d$ is under- or over-estimated, neither CREATOR nor any other related CRL algorithm will succeed in identifying the mixing functions, latent features or the DAG. When the estimated intrinsic dimension is not the same as the true value of $d$, standard metrics for evaluating how close the estimates are to their truth are not even well defined. We need to further introduce appropriate metrics that allow $d$ to be inconsistently estimated, which in our opinion is beyond the scope of our paper. In fact, even for simpler problems such as ICA (or even factor analysis), to our knowledge, the theoretical results also rely on the precise knowledge of $d$. But we agree with the reviewer that relaxing this requirement is an interesting and practically valuable problem to investigate and we will include it when we discuss possible future directions.
> ## In noisy finite samples, do you use singular-value thresholds or bootstrap ranks for deciding the 1-rank drop?
> We use singular-value thresholds and we will clarify this in Appendix C in the revision.  Below, we provide the performance of our algorithm for different choices of the threshold for $d = 5$ as a sensitivity analysis. As can be seen, the performance is insensitive to different thresholds.
>
> ### Algorithm performance for $K=d$ and $d=5$ with different threshold $\tau$
>
> | $\tau$ | 0.1  | 0.4  | 0.7  | 1.0  | 1.3  | 1.6  | 1.9  | 2.2  | 2.5  | 2.8  |
> |--------|------|------|------|------|------|------|------|------|------|------|
> | SHD    | 3.1  | 2.5  | 2.5  | 2.7  | 2.8  | 3    | 3.3  | 3.5  | 2.8  | 3    |
> | $\mathrm{LocR}^2$ | 0.69 | 0.68 | 0.72 | 0.74 | 0.70 | 0.67 | 0.71 | 0.73 | 0.71 | 0.71 |
>
>
> ### Algorithm performance for $K=2d$ and $d=5$ with different threshold $\tau$
>
> | $\tau$ | 0.1  | 0.4  | 0.7  | 1.0  | 1.3  | 1.6  | 1.9  | 2.2  | 2.5  | 2.8  |
> |--------|------|------|------|------|------|------|------|------|------|------|
> | SHD    | 3.5  | 2.5  | 2.2  | 2.0  | 2.0  | 2.0  | 2.1  | 2.7  | 2.8  | 3.2  |
> | $\mathrm{LocR}^2$ | 0.75 | 0.77 | 0.84 | 0.87 | 0.81 | 0.81 | 0.82 | 0.77 | 0.78 | 0.74 |

---

> > ### Comment · Area_Chair_LBcr · 2025-11-28
> >
> > Dear Reviewer,
> >
> > Please make sure you read the authors' response and engage with them in the discussion before the end of the discussion period on **Dec 03 '25 09:00 PM UTC**. This is a hard deadline.
> >
> > Thank you for supporting quality peer review at ICLR.
> >
> > AC

---

### Author Response · Authors · 2025-12-03
**Summary comment**

Dear Reviewers and ACs,

We would like to express our sincere gratitude for your time and for the constructive feedback on our draft. We thank all three reviewers for their generally positive comments, including Reviewer xFXr's additional feedback, the agreement that we have addressed the concerns, and the decision to **raise the score to 6**. We have thoroughly incorporated your valuable suggestions, which we believe have significantly enhanced the clarity and quality of our work.

We first briefly summarize the main contribution of our paper. Our work focuses on the stylized linear causal representation learning (CRL) using data from general environments. We design a novel CRL algorithm that can **provably identify** latent features and their causal mechanisms—up to a well-defined equivalence class—under **weaker distributional assumptions on the exogenous noise variables than the state-of-the-art literature** (also agreed by reviewers) in this setup. To evaluate the proposed method, we conduct experiments on synthetic data and apply our algorithm to uncover latent causal features in the outputs of large language models (LLMs). The results demonstrate the potential practical utility of our approach in advancing the interpretability of LLMs.

The major revisions based on reviewers' comments are summarized as follows:

- Exploring downstream tasks constitutes an essential area of research in CRL and serves as a key evaluation metric. Consequently, we have included a discussion on potential directions for downstream applications as part of the future work in the last section of the main text.

- To make our draft easier to follow, we will provide toy examples in Appendix G to illustrate the frequently used notations including $\sim_\pi,\ \sim_\triangle$ and $\sim_\mathrm{sur}$ and the main steps of our algorithm.

- In Introduction section, we add a discussion comparing our work with [1,2] to clarify that, while related, our research addresses a distinct problem: whereas [1,2] primarily aim to identify the causal structure among both observed and latent variables, our work focuses on identifying the latent variables themselves and the causal graph solely from observational data.

- To clarify the inconsistency in line 193 such that “the latent features and causal DAG can be uniquely recovered.”, we will change line 191-193 to “Consistent with much of the current CRL literature, we focus on identifiability -- namely, whether the latent features and causal DAG can be recovered up to $\sim_{\mathrm{sur}}$ in the limit of infinite data, and whether our proposed procedure achieves this recovery in theory.”

- We clarify the computation of the rank in Algorithm 2 in Appendix C.
- We provide a comparison between our pruning subroutine and the Identify-Parents algorithm in Jin \& Syrgkanis (2024) and show why our algorithm is more efficient as a remark in Appendix C.

Taken together, we hope that we have addressed the reviewers' comments and our paper is in a better shape to be published in ICLR.

Best regards,

Authors

__Reference__

[1] A Versatile Causal Discovery Framework to Allow Causally-Related Hidden Variables. Xinshuai Dong, Biwei Huang, Ignavier Ng, Xiangchen Song, Yujia Zheng, Songyao Jin, Roberto Legaspi, Peter Spirtes, Kun Zhang.

[2] Generalized Independent Noise Condition for Estimating Causal Structure with Latent Variables. Feng Xie, Biwei Huang, Zhengming Chen, Ruichu Cai, Clark Glymour, Zhi Geng, Kun Zhang.

---

### Meta-Review · Area_Chair_J4dn · 2026-01-13

**Summary:**

This paper proposes a new method linear CRL called CREATOR. Like previous works, it relies on LiNGAM-style assumptions for identification, which substantially simplifies the analysis. A common theme in the reviews was skepticism that there is a meaningful improvement over prior assumptions.

**Reviewer Concerns:**

- rpXZ Limited novelty: **No**
- rpXZ Over/under estimation: **No**
- rpXZ Noisy finite samples: Yes
- xFXr Assumption relaxation and identifiability: **No**
- xFXr ICA and independence testing: **No**
- xFXr Fragility of pruning and disentanglement: **No**
- xFXr Empirical limitations: **No**
- xFXr Presentation and clarity: Yes
- ba3u Linearity: **No**
- ba3u Same observed variables: **No**
- ba3u Dense writing: **No**
- ba3u Missing citations: Yes
- ba3u Efficiency: **No**

**Reviewer Scores:**

rpXZ 6->6
xFXr 4->5
ba3u 6->6

---

### Decision · Program_Chairs · 2026-01-26

Reject